# Mechanosensory neurons control the timing of spinal microcircuit selection during locomotion

Steven Knafo[1,2†], Kevin Fidelin[1†], Andrew Prendergast[1], Po-En Brian Tseng[1], Alexandre Parrin[1], Charles Dickey[1], Urs Lucas Böhm[1], Sophie Nunes Figueiredo[1], Olivier Thouvenin[1], Hugues Pascal-Moussellard[1,2], Claire Wyart[1*]

[1]Institut du Cerveau et de la Moelle épinière (I.C.M.), Sorbonne Universités, UPMC Univ Paris 06, Inserm, CNRS, Institut du Cerveau et la Moelle épinière, Hôpital Pitié-Salpêtrière, Paris, France; [2]AP-HP, Hôpital Pitié-Salpêtrière, Paris, France

**Abstract** Despite numerous physiological studies about reflexes in the spinal cord, the contribution of mechanosensory feedback to active locomotion and the nature of underlying spinal circuits remains elusive. Here we investigate how mechanosensory feedback shapes active locomotion in a genetic model organism exhibiting simple locomotion—the zebrafish larva. We show that mechanosensory feedback enhances the recruitment of motor pools during active locomotion. Furthermore, we demonstrate that inputs from mechanosensory neurons increase locomotor speed by prolonging fast swimming at the expense of slow swimming during stereotyped acoustic escape responses. This effect could be mediated by distinct mechanosensory neurons. In the spinal cord, we show that connections compatible with monosynaptic inputs from mechanosensory Rohon-Beard neurons onto ipsilateral V2a interneurons selectively recruited at high speed can contribute to the observed enhancement of speed. Altogether, our study reveals the basic principles and a circuit diagram enabling speed modulation by mechanosensory feedback in the vertebrate spinal cord.

*For correspondence: claire. wyart@icm-institute.org

†These authors contributed equally to this work

Competing interests: The authors declare that no competing interests exist.

## Introduction

The generation of complex motor behaviors at a given speed relies on the sequential selection and orderly activation of groups of muscles by the nervous system (*Bellardita and Kiehn, 2015*). At the level of locomotor circuits in the spinal cord, premotor excitatory V2a interneurons and motor neurons are recruited in a frequency-dependent manner (*Crone et al., 2008*; *Kimura et al., 2006*; *McLean et al., 2007*), a feature thought to underlie speed control in zebrafish (*McLean et al., 2008*). Recently, circuit-mapping experiments revealed preferential connectivity profiles between identified groups of V2a interneurons and motor neurons suggesting that distinct microcircuit modules could regulate the mobilization of individual muscles during locomotion (*Ampatzis et al., 2014*; *Bagnall and McLean, 2014*). Genetic ablation of V2a interneurons in mammals is associated with deficits in left-right coordination and gait transitions suggesting that V2a interneurons play a key role in controlling speed across vertebrates (*Crone et al., 2008*). The nature of spinal circuits involved in motor pattern generation and speed control has mainly been studied during fictive locomotion when sensory feedback is absent (*Kiehn, 2016*; *McLean and Dougherty, 2015*). However, less is known about the cellular and circuit mechanisms regulating the timing of microcircuit selection in the spinal cord during movement, which incorporate proprioceptive feedback from muscle contraction.

Peripheral mechanosensory feedback plays a critical role in setting the timing of muscle activation during movement (*Windhorst, 2007*). In particular, stimulation of afferent sensory neurons in decerebrate and paralyzed cats can reorganize the pattern of motor output during fictive locomotion, either by entraining or resetting the locomotor rhythm, depending on the nature of active spinal circuits (*Conway et al., 1987*; *Guertin et al., 1995*; *Jankowska, 1992*; *McCrea, 2001*; *Rossignol et al., 2006*). In addition, altering the development of peripheral sensory neurons leads to dramatic perturbations of motor sequences during walking and swimming in mice, suggesting that mechanosensory circuits can directly control the activity of spinal central pattern generators responsible for rhythmic motor outputs (*Akay et al., 2014*; *Takeoka et al., 2014*). However, the molecular identity of spinal neurons mediating these effects in mammals remains elusive.

To understand the contribution of mechanosensory feedback to speed control, we investigated the role of spinal mechanosensory neurons during acoustic escape responses in the zebrafish larva, a genetic model system with tractable locomotor behaviors that can be analyzed using fine kinematic analysis. We developed a bioluminescence-based assay to monitor the activity of populations of spinal motor neurons in moving animals and combined this approach with calcium imaging and quantitative kinematic analysis of locomotor behaviors to compare the dynamics of motor activity in active and fictive swimming. We demonstrate that glutamatergic mechanosensory neurons enhance the recruitment of spinal motor neurons during motion by increasing locomotor frequency, and consequently, the speed of locomotion. In particular, we show that mechanosensory feedback selectively enhances the fast regime over the slow regime during escape responses, thereby promoting the transition between fast and slow locomotion. Using optogenetics for circuit connectivity mapping, we identified a functional microcircuit module formed by mechanosensory Rohon-Beard neurons onto ipsilateral glutamatergic V2a interneurons selectively active during fast locomotion. Altogether our results reveal how spinal microcircuits controlling locomotor frequency can be selected by mechanosensory pathways to enhance speed during active locomotion.

## Results

### Monitoring the recruitment of spinal motor neurons in moving animals

In order to investigate the contribution of sensorimotor circuits during locomotor activity in zebrafish larvae, we used the bioluminescent calcium sensor GFP-Aequorin to monitor calcium activity emitted by identified neuronal populations in freely moving animals (*Baubet et al., 2000*; *Naumann et al., 2010*; *Shimomura et al., 1962*). To maximize expression of GFP-Aequorin in zebrafish motor circuits, we codon-optimized the Aequorin sequence and targeted its expression to spinal motor neurons. We recorded bioluminescence signals emitted from spinal motor neurons during active behaviors elicited by an acoustic stimulus in *Tg(mnx1:gal4;UAS:GFP-aequorin-opt)* zebrafish larvae at four days post-fertilization (dpf) (*Figure 1A–B*, see Materials and methods). To remove the concern of collecting unspecific signals following large changes in intracellular calcium concentration in muscle fibers during tail contractions, we systematically determined both in live animals and after GFP immunostainings that there was no expression in non-neuronal tissues such as muscles (*Figure 1B*, n = 5 larvae, *Video 1*). Acoustic stimulations triggered stereotypical escape responses characterized by an asymmetrical initial C-bend followed by fast swimming. In rare cases, possibly due to coelenterazine application, acoustic stimulations led to stereotypical slow swims characterized by symmetrical bends with undulations propagating from rostral to caudal along the tail (7.4% of trials, 21 out of 283 stimulations; see *Mirat et al. [2013]*) (*Figure 1C–D*, *Videos 2–3*, n = 10 larvae, see *Budick and O'Malley, 2000*). Each locomotor behavior was associated with a specific bioluminescence signal (*Figure 1D*, *Videos 2–3*). While time-to-peak and time decay of bioluminescence signals were invariant across maneuvers (data *not shown*), stereotypical escape responses generated larger bioluminescence amplitudes than slow swims (*Figure 1D–E*, mean bioluminescence amplitude = 30.8 + /- 1.4 photons/10 ms for escapes and 4.1 ± 1.6 photons/10 ms for slow swimming, p<0.001). Bioluminescence signal amplitude correlated with the maximal angle of the tail bend (*Figure 1F*, R = 0.4, p<0.001), suggesting that spinal motor neuron recruitment is increased during behaviors with larger tail bends and faster locomotor frequencies. However, monitoring global bioluminescence signals lacks the single cell resolution required to explain whether these increased bioluminescence signals result from the recruitment of additional motor neurons

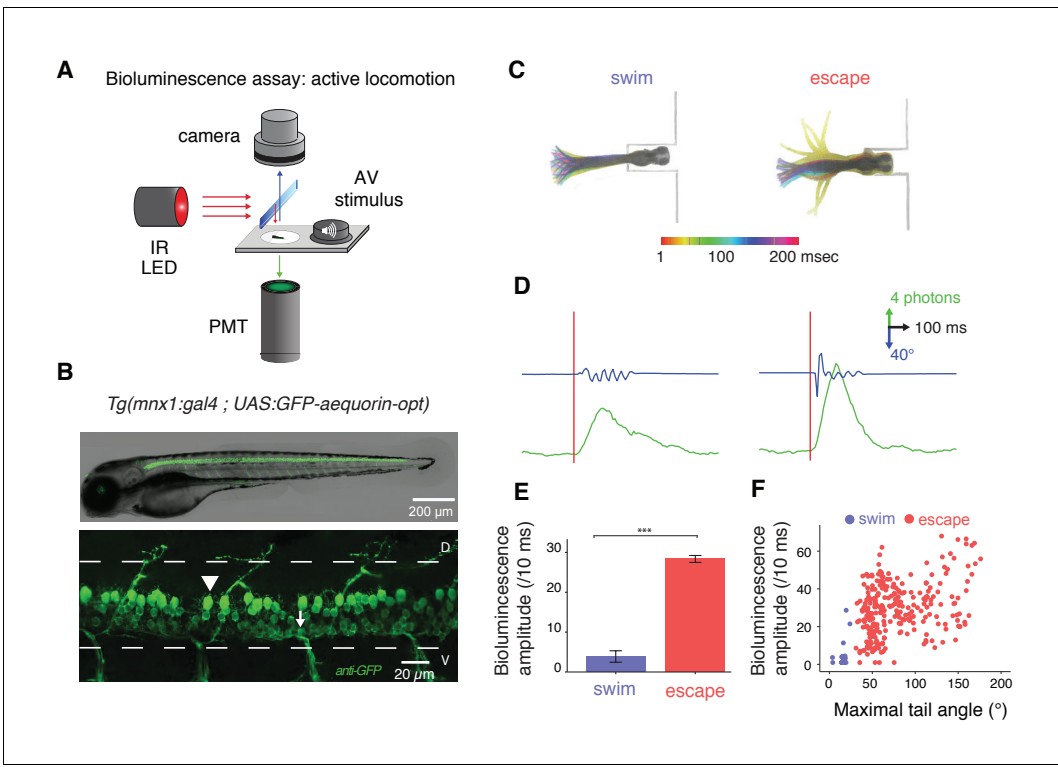

**Figure 1.** Amplitude of bioluminescence signals in spinal motor neurons correlates with the type of locomotor maneuver during active swimming. (**A**) Bioluminescence signals emitted from spinal motor neurons in *Tg(mnx1: gal4;UAS:GFP-aequorin-opt)* zebrafish larvae at 4 dpf were recorded using a photomultiplier tube under infrared illumination during active behaviors elicited by an acoustic stimulus. (**B**) Live fluorescent image (upper panel) and immunohistochemistry for GFP (lower panel) in a 4 dpf *Tg(mnx1:gal4;UAS:GFP-aequorin-opt)* zebrafish larva showing selective expression in spinal motor neurons (arrowhead: dorsal primary, arrow: ventral secondary motor neurons), and strictly no expression in muscle fibers (n = 5). (**C**) Motor behaviors elicited by acoustic stimuli. Superimposed traces illustrate the amplitude of tail contractions over time, for each behavior. Traces are color-coded according to the delay from stimulus onset. Automated categorization classified maneuvers into escapes (n = 245/283) or swims (n = 21/283) (n = 10 larvae and 300 trials). (**D**) Example traces of typical bioluminescence signals and kinematic parameters observed for each category. (**E**) Mean bioluminescence amplitude was higher for escapes (28.4 ± 0.9 photons/10 ms; normalized amplitude per larva = 0.41 + /- 0.18) than swims (3.9 ± 1.4 photons/10 ms, p<0.001; normalized amplitude = 0.06 + /- 0.02, p<0.001). (**F**) Correlation between bioluminescence signal amplitude and maximum tail angle amplitude (R = 0.4, p<0.001).

during fast locomotion or from increased signals in the same pool of active cells. To decipher between these possibilities we recorded the activity of pools of motor neurons using the genetically-encoded calcium indicator GCaMP6f

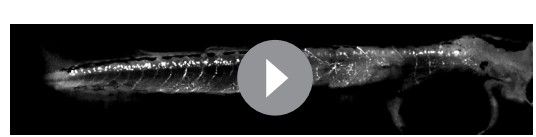

**Video 1.** Expression of GFP in a 4 dpf *Tg(mnx1:gal4; UAS:GFP-aequorin-opt)* larva revealed by immunohistochemistry (one example shown out of the n = 5 larvae where IHC was performed).

**Video 2.** Bioluminescence signal emitted by a 4 dpf *Tg (mnx1:gal4; UAS:GFP-aequorin-opt)* free-tailed transgenic larva and corresponding kinematics during escape.

during spontaneous fictive slow swimming and elicited fictive escape responses in *Tg(mnx1:gal4; UAS:GCaMP6f,cryaa:mCherry)* (*Figure 2A–B*, *Böhm et al., 2016*). This strategy allowed us to monitor dynamics of both individual neurons and populations of neurons during locomotor behaviors previously probed using bioluminescence. At 4 dpf, fictive locomotor burst frequencies ranged between 20–30 Hz for slow swims (left panel of *Figure 2C*, see *Masino and Fetcho [2005]*) and 20–80 Hz for escapes (right panel of *Figure 2C*). During slow swimming, a small fraction of ventral motor neurons was active (23.2%, left panel of *Figure 2D*, *Figure 2E*, *Video 4*). In contrast, the majority of motor neurons, including large dorsal motor neurons, was recruited during escapes (88.4%, right panel of *Figure 2D*, *Figure 2E*, and *Video 5*). Dorsalmost motor neurons showed larger calcium transients than ventral ones (*Figure 2D*), and the averaged ΔF/F across all recruited motor neurons was higher during escapes compared to slow swims (*Figure 2F*, mean ΔF/F/active cell = 91.1 + /- 4.7% during escape responses and 25.9 ± 3.8% during slow swimming, p<0.001, n = 10 larvae). These data revealed that the generation of fast escape responses requires the recruitment of more spinal motor neurons compared to slow swimming and that the amplitude of calcium signals are higher in cells active during escapes compared to spontaneous swimming. Therefore, we conclude that high bioluminescence signal amplitude during escape responses compared to slow swims likely results from the combination of two effects: a larger pool of active cells driven by an increase in swimming frequency and larger calcium transients per active neuron.

## Mechanosensory feedback enhances the recruitment of spinal motor neurons

To test whether mechanosensory feedback modulates the global recruitment of spinal motor neurons during locomotion, we compared motor dynamics in active and fictive escape responses by recording bioluminescence before and after blocking muscle contraction following the bath application of cholinergic blocker pancuronium bromide (*Figure 3A*, *Panier et al., 2013*). Across all *Tg (mnx1:gal4;UAS:GFP-aequorin-opt)* larvae, mean bioluminescence amplitude was markedly decreased in paralyzed compared to motile animals (*Figure 3B–C*, signal amplitude before paralysis = 26.6 + /- 0.9 photons/10 ms; signal amplitude after paralysis = 11.1 + /- 0.4 photons/10 ms, p<0.001, n = 10 larvae). Within each larva, the mean normalized bioluminescence amplitude was decreased seven-fold (0.056 ± 0.08 versus 0.36 ± 0.18, p<0.001). Since this effect could be partly explained by the inhibition of acetylcholine receptors located on targets of reticulospinal neurons, such as the Mauthner cell (*Koyama et al., 2011*), we conducted the same experiments in immotile *relaxed* (*cacnb1$^{ts25}$*) mutant zebrafish (*Granato et al., 1996*; *Zhou et al., 2006*). The mean bioluminescence amplitude was markedly decreased in immotile *Tg(cacnb1$^{ts25/ts25}$; mnx1:gal4; UAS:GFP-aequorin-opt)* larvae compared to motile *Tg(mnx1:gal4, UAS:GFP-aequorin-opt)* control siblings (*Figure 3D–E*, mean signal amplitude in motile siblings = 37.6 + /- 1.4 photons/10 ms; mean signal amplitude in immotile mutants = 9.8 + /- 0.4 photons/10 ms, p<0.001, n = 3 larvae). Altogether these results demonstrate that the recruitment of spinal motor neurons is enhanced during active locomotion, indicating that mechanosensory feedback pathways recruited during muscle contraction contributes to increasing spinal motor output. However, from these measurements it is unclear whether the decrease in signal amplitude observed in Aequorin-expressing larvae when muscle contraction is impaired is indicative of a reduction in amplitude or frequency of motor output during acoustic escapes responses.

## Silencing mechanosensory neurons decreases the speed of locomotion

To decipher between these possibilities, we probed the contribution of mechanosensory neurons during active swimming through fine kinematic analysis of locomotor behaviors in freely moving animals. We expressed the zebrafish-optimized Botulinum toxin under the control of the *isl2b* promoter (*Auer et al., 2015*; *Böhm et al., 2016*; *Sternberg et al., 2016*) (see Materials and methods and *Video 6*) in order to block synaptic release in glutamatergic mechanosensory neurons, namely Rohon-Beard neurons (RB) and dorsal root ganglia (DRG) in the spinal cord and trigeminal ganglia (*Figure 4A*, note the bundle of RB and DRG axons ascending in the spinal cord in *Figure 4B*, (*Won et al., 2011*)). We performed quantitative kinematic analysis of locomotor behaviors in freely-swimming *Tg(isl2b:gal4,cmlc2:eGFP;UAS:BoTxBLC-GFP)* larvae and found alteration of several kinematic parameters compared to control siblings (*Figure 4C–D*, n = 304 larvae, 890 escapes). Distance

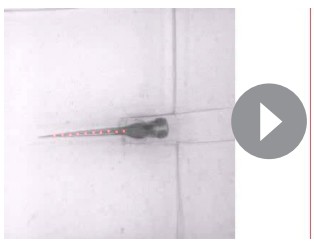

**Video 3.** Bioluminescence signal emitted by a 4 dpf *Tg (mnx1:gal4; UAS:GFP-aequorin-opt)* free-tailed transgenic larva and corresponding kinematics during swimming.

traveled (*Figure 4E1*) and total number of oscillations (*Figure 4E2*) were unaffected in animals deprived of mechanosensory feedback. However, we noted that tail-beat frequency (TBF) (*Figure 4E3*) and, accordingly, speed (*Figure 4E4*) were markedly reduced in BoTxBLC-GFP[+] animals. Coincident with the reduction in TBF and stability of number of oscillations, the escape duration increased in larvae deprived of mechanosensory feedback (*Figure 4E5*, from 168.1 ± 2.4 ms to 182.0 ± 3 ms). While the amplitude of the first C-bend was not affected in BoTxBLC-GFP[+] animals (*Figure 4—figure supplement 1*), the amplitudes of subsequent tail bends were increased. This was especially pronounced on the ipsilateral side, suggesting a modulation of motor circuits recruited after the response onset. However, from this

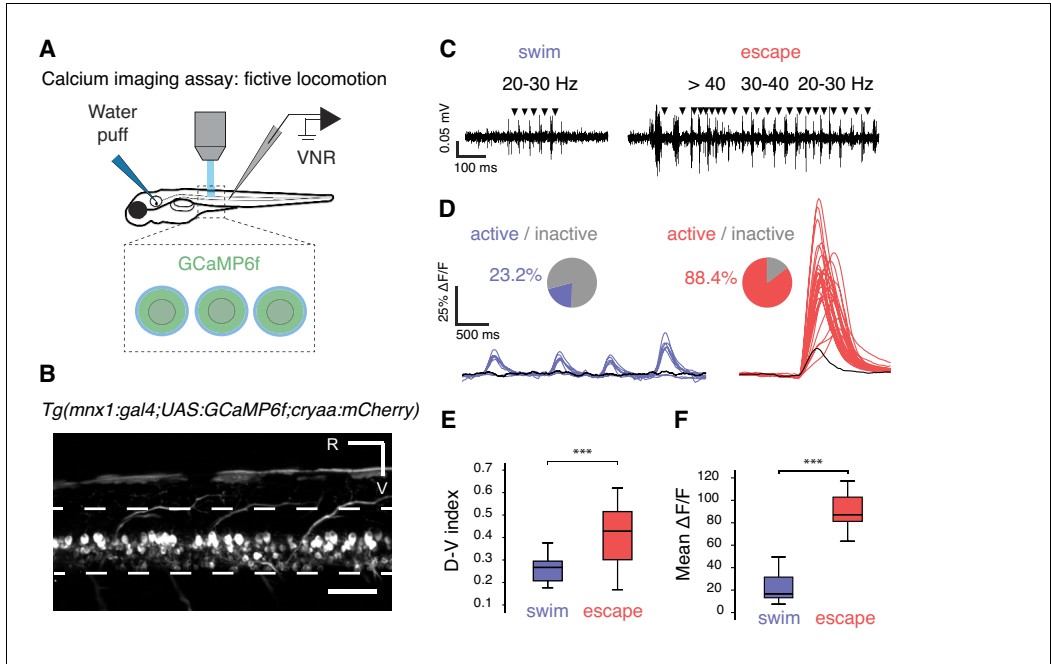

**Figure 2.** Calcium imaging in paralyzed larvae confirms a larger recruitment of spinal motor neurons for fast escapes than for slow swimming. (A) Calcium transients emitted from spinal motor neurons in *Tg(mnx1:gal4;UAS: GCaMP6f,cryaa:mCherry)* zebrafish larvae at 4 dpf were recorded simultaneously with ventral nerve root recordings (VNR) during fictive behaviors elicited by a water puff to the otic vesicle. (B) Expression pattern in a *Tg(mnx1:gal4; UAS:GCaMP6f,cryaa:mCherry)* larva at 4 dpf (R is rostral, V is ventral; scale bar is 50 μm). (C) VNR for each fictive behavior illustrates typical spontaneous fictive slow swims and induced fictive escapes: burst frequencies ranged between 20–30 Hz for slow swims (left panel) and 20–80 Hz for escapes (right panel). (D) GCaMP6f signals from individual motor neurons during spontaneous slow swimming (4 swims, left panel) and during an evoked escape response (right panel). For each recording, out-of-focus light was estimated within the spinal cord (black trace) and used as a criterion to determine the active vs inactive status of each cell. Pie charts represent the proportion of active cells in each behavior: 16/69 cells across 27 swims versus 61/69 cells across 12 escapes (n = 3 larvae). (E) Dorso-ventral (D–V) position of cells recruited during each maneuver shows dorsal motor neurons only active during escapes (the dorso-ventral axis within the spinal cord is normalized to 0 at the ventral limit and 1 the dorsal limit; mean D-V position for escapes = 0.41 + /- 0.02 versus 0.25 ± 0.01, p<0.001, n = 78 cells in n = 3 larvae). (F) Mean ΔF/F amplitude was higher during escapes compared to spontaneous swims across larvae (91.2 ± 4.7% versus 25.9 ± 3.8%, p<0.001, 12 escapes and 27 swims in n = 3 larvae).

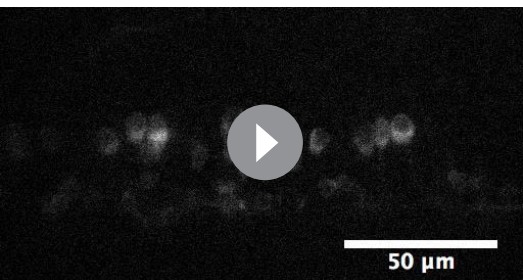

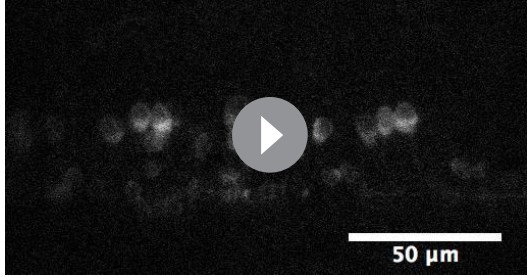

**Video4.** Calcium imaging of spinal motor neurons in a 4 dpf *Tg(mnx1:gal4; UAS:GCaMP6f)* larva during fictive slow swimming. Animals were paralyzed with bungarotoxin injections. A small fraction of ventral motor neurons is active. Recordings were acquired at 20 Hz. Movies were sped up by 2.5 X.

**Video 5.** Calcium imaging of spinal motor neurons in a 4 dpf *Tg(mnx1:gal4; UAS:GCaMP6f)* larva during a fictive escape response. Animals were paralyzed with bungarotoxin injections. A large portion of both ventral and dorsal motor neurons is recruited. Recordings were acquired at 20 Hz. Movies were sped up by 2.5 X.

global analysis it is unclear if the effects on locomotor frequency and speed by mechanosensory feedback reflect a specific modulation of fast and/or slow swimming.

To determine if mechanosensory feedback selectively modulates fast versus slow locomotion in our silencing experiments, we separated the fast from the slow locomotor regime in elicited escape responses by applying a cutoff at 30 Hz, a threshold that defines the recruitment of fast motor neurons during swimming (*McLean et al., 2008*; *Severi et al., 2014*) (*Figure 5A–B*). This analysis revealed a decrease of the distance travelled, duration, number of cycles, speed, and tail-beat frequency of the fast component in BoTxBLC-GFP+ compared to wild type siblings (*Figure 5C1–C5*). In contrast, there was an increase of the distance travelled, duration and number of cycles of the slow component in BoTxBLC-GFP+ while the tail-beat frequency and speed were not altered (*Figure 5D1–D5*). Remarkably, the total number of cycles remains constant in BoTxBLC-GFP+

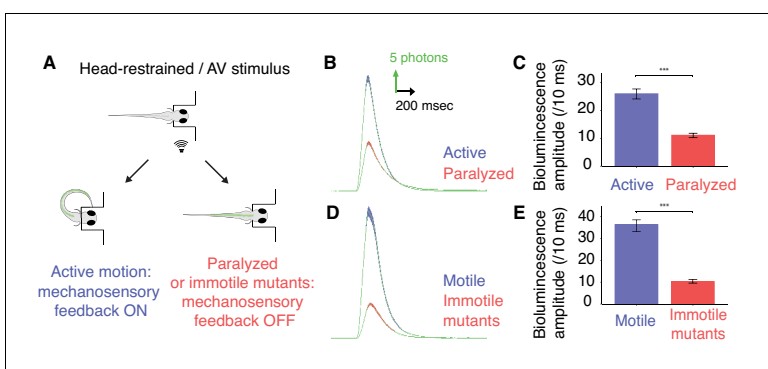

**Figure 3.** Mechanosensory feedback enhances the recruitment of spinal motor neurons during active locomotion. (A) To compare the recruitment of spinal motor neurons when mechanosensory feedback was present ('active locomotion') or suppressed ('fictive locomotion'), we conducted bioluminescence assays in 4 dpf *Tg(mnx1:gal4; UAS:GFP-aequorin-opt)* zebrafish larvae before and after paralysis using pancuronium bromide, and in immotile 4 dpf *Tg(mnx1:gal4;UAS:GFP-aequorin-opt;cacnb1^{ts25/ts25})* mutants compared to their motile siblings. (B) Bioluminescence signals from motor neurons revealed a marked decrease in bioluminescence amplitude after paralysis. (C) Quantification of the change in mean bioluminescence amplitude (before paralysis: 26.6 ± 0.9 photons/10 ms; after paralysis: 11.1 ± 0.4 photons/10 ms, n = 10 larvae in each group, 30 trials per larva, p<0.001). (D) Similarly, averaged bioluminescence signals from motor neurons were markedly decreased in immotile *Tg(mnx1:gal4;UAS:GFP-aequorin-opt;cacnb1^{ts25/ts25},)* mutant larvae when compared with motile siblings. (E) Mean bioluminescence amplitude in motile siblings (37.6 ± 1.4 photons/10 ms) compared to immotile mutants (9.8 ± 0.4 photons/10 ms, n = 300 trials in 10 larvae for each group, p<0.001).

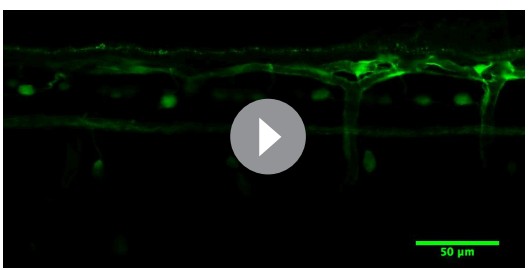

**Video 6.** Expression of GFP in a 4 dpf *Tg(isl2b:gal4, cmcl2:eGFP;UAS:GFP-aequorin-opt)* larva revealed by immunohistochemistry (one example shown out of the n = 4 larvae where IHC was performed).

compared to wild type siblings (*Figure 4E2*), indicating that the decrease of fast swimming frequencies observed in BoTxBLC-GFP[+] larvae leads to a decrease of the fast component duration and induces an increase of the slow component duration. Therefore, the reduction of speed and locomotor frequency observed in mechanosensory-deprived animal is specifically restricted to the fast component but not the slow one (*Figure 5A–B*). Altogether, our results demonstrate that mechanosensory feedback enhances speed of locomotion by promoting the transition between fast and slow locomotion during escape responses. Due to the fact that the Gal4/UAS combinatorial expression system labels cells in a variegated manner in zebrafish, we only observed BoTxBLC-GFP in half the RB neurons although the *isl2b* promoter drives expression in all RB neurons, dorsal root ganglia and trigeminal neurons. This observation suggests that the effects of silencing mechanosensory feedback are likely underestimated in our behavior experiments.

## Mechanosensory neurons synapse onto V2a interneurons involved in fast locomotion

Given the selective effect on tail-beat frequency and speed during the fast component of escape responses following the silencing of mechanosensory feedback, we reasoned that mechanosensory neurons may be able to selectively recruit spinal interneurons involved in fast locomotion. Previous studies in zebrafish larvae showed that spinal V2a interneurons are recruited in a topographic and frequency-dependent manner during fictive locomotion, with dorsalmost V2a interneurons solely active during fast locomotion (*McLean et al., 2008*, *2007*). Ablation, silencing and optogenetic manipulations revealed that these neurons constitute an important class of excitatory interneurons driving the recruitment of motor neurons—especially during fast locomotion (*Ampatzis et al., 2014*; *Bagnall and McLean, 2014*; *Crone et al., 2008*; *Kimura et al., 2006*, *2013*; *Ljunggren et al., 2014*; *McLean et al., 2007*). Interestingly, we noticed patterns of perisomatic innervation of dorsal V2a cell bodies by axonal projections originating from one cell type targeted by the *isl2b* driver, Rohon-Beard cells (*Figure 6A–D*, mean soma position of V2a = 0.72 + - 0.04, n = 12 cells in 4 larvae).

In order to test whether Rohon-Beard neurons form functional synapses onto V2a interneurons, we performed optogenetic-mediated mapping of synaptic connectivity in the spinal cord (*Figure 6D*, *Figure 7D*). Rohon-Beard cells are low input resistance neurons and required current injections ranging from 200 to 700 pA to generate single spikes (*Figure 7B*, n = 6 cells, 250 ms injection steps). In comparison, dorsalmost V2a neurons only required from 15 to 50 pA to fire action potentials (*Figure 7H*, n = 5 cells). As a consequence, RBs might not be amenable to optogenetic stimulations at larval stages using first generation opsins such as ChR2-H134R (*Boyden et al., 2005*). To circumvent this potential issue we expressed the novel opsin CoChR-tdTomato (*Klapoetke et al., 2014*) in Rohon Beard cells and tested their responses to blue light stimulation (*Figure 6D*, *Figure 7A*). We found that 1 to 5 ms blue light pulses were sufficient to elicit photocurrents driving single action potentials in Rohon-Beard neurons at 4 and 5 dpf (*Figure 7C and G*, mean time to peak from the onset of the light pulse = 2.80 + /- 0.29 ms, n = 6 cells). We then recorded from target V2a interneurons while photostimulating RB neurons in *Tg(isl2b:gal4,cmcl2:eGFP;UAS:CoChR-tdTomato;chx10:DsRed)* larvae (*Figure 7D–F*). We found that RB stimulation led to short latency excitatory post-synaptic currents (EPSCs) in target V2a interneurons (*Figure 7D–E and G*, mean delay from the onset of light pulse = 3.71 + /- 0.19 ms, n = 8 cells in 8 larvae). To assess whether RB-V2A ipsilateral connection might be monosynaptic, we analyzed the lag between RB spikes time-to-peak and V2a EPSCs onset and found that it was below 1 ms (*Figure 7G*, 0.9 ms), a value compatible with monosynaptic connection (see *Li et al. [2007a]*) where cutoff is set at 3 ms for monosynaptic connections. Assuming a synaptic delay of 0.5 ms, the estimated conduction velocity along the axon of RB neuron would be 0.75 m/s, a value compatible with previous measurement in the field (*Li et al., 2007a*, *2004a*). EPSC amplitude induced by RB stimulation in V2a interneurons were on

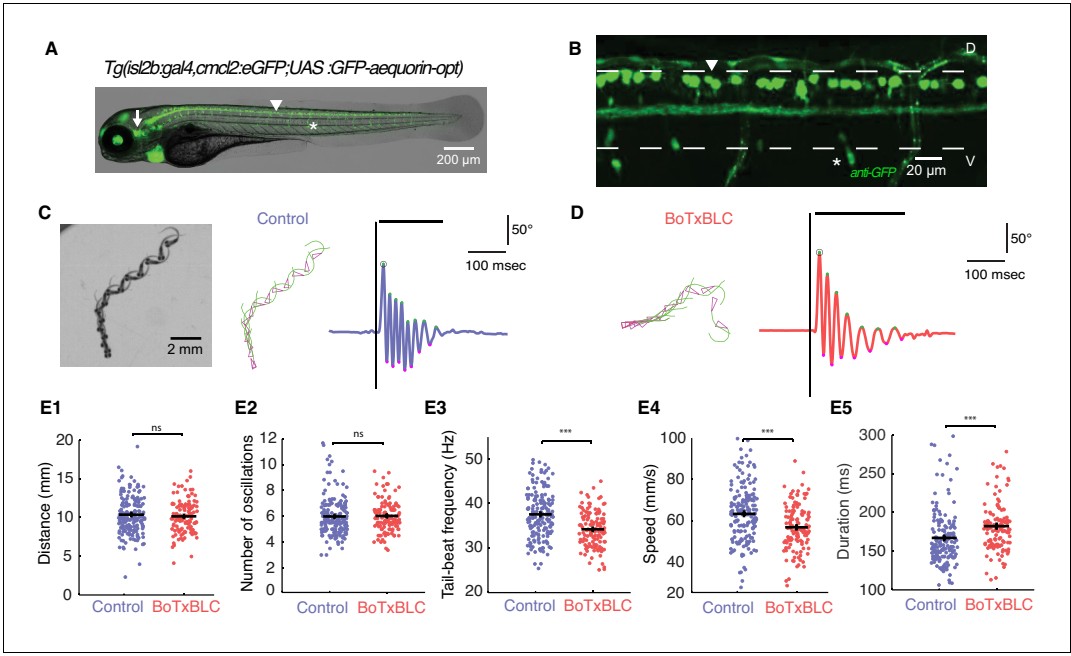

**Figure 4.** Silencing mechanosensory neurons decreases speed of locomotion. (**A**) In vivo fluorescence image and (**B**) immunohistochemistry for GFP in 4 dpf *Tg(isl2b:gal4,cmcl2:eGFP;UAS:GFP-aequorin-opt)* double transgenic zebrafish larva show selective expression of GFP-aequorin in mechanosensory neurons: trigeminal ganglia (arrow), Rohon-Beard spinal neurons and their ascending axons (arrowhead) and dorsal root ganglia (*), but no expression in muscle fibers (n = 4). (**C–D**) Superimposed images from the raw high-speed camera recording (left), the corresponding automated tracking (middle, pink triangles represent the head and green lines the tail of the fish) and tail angle over time extracted from the tracking analysis (right; stimulus is represented by the black vertical line, duration of the event is the black horizontal line) of a typical escape elicited by an acoustic stimulus in a freely-swimming control sibling larva at 6 dpf (**C**), and in a *Tg(isl2b:gal4,cmcl2:eGFP;UAS:BoTxBLC-GFP)* larva (**D**). (**E1**) Distance travelled was unchanged in BoTxBLC$^+$ and control larvae (10.2 ± 0.2 mm versus 10.6 ± 0.2 mm, p=0.1). (**E2**) Number of oscillations was unchanged in BoTxBLC$^+$ and control larvae (6.0 ± 0.2 versus n = 6.1 + /- 0.2 oscillations, p=0.6). (**E3**) BoTxBLC$^+$ larvae showed a decreased tail-beat frequency (TBF, 34.2 ± 0.4 Hz versus 37.8 ± 0.3 Hz, p<0.001). (**E4**) BoTxBLC$^+$ larvae showed a decreased speed of escape responses (56.9 ± 1.1 versus 64.1 ± 0.9 mm/s, p<0.001). (**E5**) Escape duration was increased in BoTxBLC$^+$ larvae compared to control siblings (182.0 ± 3.0 ms versus 168.1 ± 2.4 ms, p<0.001). (For all parameters: control group: n = 176 larvae from 8 clutches, n = 561 escapes; BoTxBLC+ group: n = 128 larvae from 8 clutches, n = 329 escapes).

The following figure supplement is available for figure 4:

**Figure supplement 1.** Silencing mechanosensory neurons alters bends amplitude only after the initial C-bend of the escape response.

---

average 28.14 ± 3.9 pA, leading to estimated depolarization of 11.71 ± 2.59 mV on target V2a (n = 8). In some cases, RB inputs were therefore sufficient to elicit action potentials in V2a interneurons tested in current clamp mode (*Figure 7F*). Given the glutamatergic nature of RB neurons, we incubated the preparation with blockers of glutamatergic neurotransmission and found that it abolished EPSCs in target V2a (*Figure 7I*, n = 2 cells). Altogether, these results indicate a monosynaptic connection between RB neurons and ipsilateral dorsalmost V2a interneurons and demonstrate the ability of RB neurons to recruit selectively a subpopulation of V2a interneurons controlling fast swimming in the dorsal spinal cord. This direct pathway from RB neurons onto ipsilateral 'fast' V2a neurons may act in concert with afferences from dorsal root ganglia and trigeminal neurons to enhance speed during active locomotion.

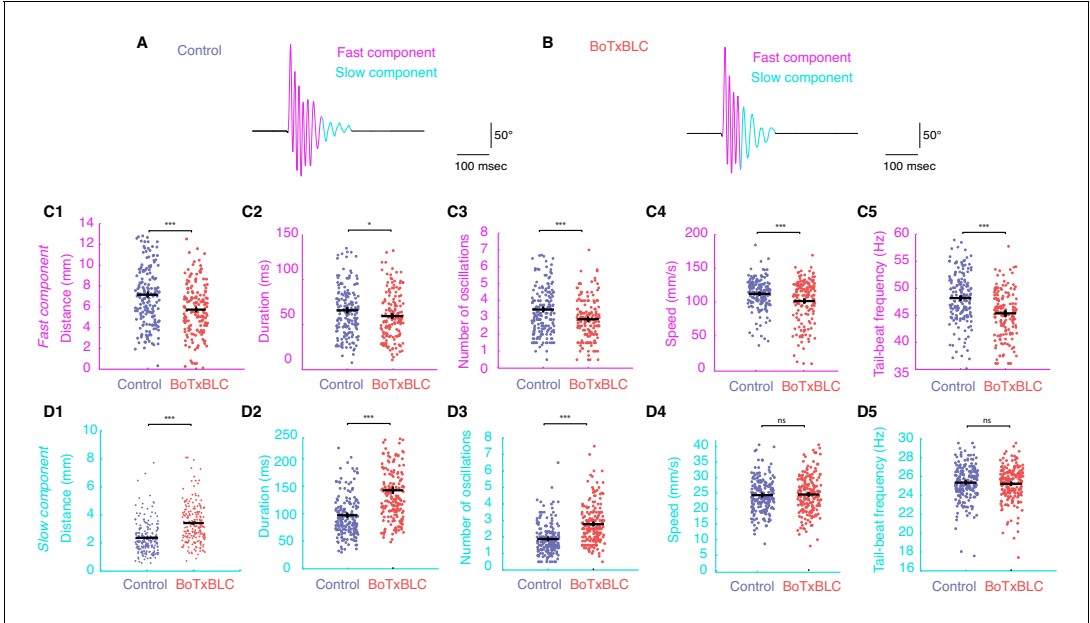

**Figure 5.** Silencing mechanosensory neurons accelerates the transition between fast and slow locomotion during acoustic escape responses. (A–B) We separately analyzed the fast component of the escape response (cycles with TBF >30 Hz) and the slow component (cycles with TBF ≤30 Hz) as shown for control (n = 176 larvae from 8 clutches, n = 561 escapes) and BoTxBLC[+] larvae (n = 128 larvae from 8 clutches, n = 329 escapes). (C1–C5) Within the fast regime of the escape, silencing mechanosensory feedback reduces the distance travelled in BoTxBLC[+] larvae compared to control siblings (C1, 5.9 ± 0.2 versus 7.4 ± 0.2 mm, p<0.001), duration (C2, 60.0 ± 2.1 versus 67.8 ± 1.9 ms, p=0.005), number of oscillations (C3, 2.9 ± 0.1 versus 3.6 ± 0.1 oscillations, p<0.001), speed (C4, 104.3 ± 1.9 versus 112.6 ± 1.6 mm/s, p=0.001), and TBF (C5, 45.8 ± 0.4 versus 48.4 ± 0.3 Hz, p<0.001). (D1–D5) Within the slow component of the escape, silencing mechanosensory feedback in BoTxBLC[+] larvae increases the distance travelled (D1, 3.5 ± 0.1 versus 2.4 ± 0.1 mm, p<0.001), duration (D2, 140.9 ± 3.4 versus 97.1 ± 3.2 ms, p<0.001, p<0.001) and number of oscillations (D3, 2.8 ± 0.1 versus 1.9 ± 0.1, p<0.001) but has no effect on speed (D4, 24.9 ± 0.4 versus 24.6 ± 0.4 mm/s, p=0.5) and tail-beat frequency (D5, 25.3 ± 0.1 versus 25.4 ± 0.1 Hz, p=0.5). For all parameters: n = 304 larvae, number of fast components = 1013; number of slow components = 1265.

## Discussion

Understanding the contribution of peripheral mechanosensory feedback during locomotion is an important question in the field of motor control. Although the state-dependent nature of spinal modulation by afferent muscle fibers has been appreciated for more than a century in limbed vertebrates, little is currently known on the role of mechanosensory feedback during ongoing locomotion. The reasons for this lie in the difficulty of recording and manipulating neural circuits in moving animals and in the fact that mechanosensory feedback is absent in fictive preparations. In this study, we took advantage of the genetic and optical accessibility of the zebrafish larva to unravel mechanisms of circuit and behavior modulation by mechanosensory feedback using a combination of optical methods for circuit interrogation during animal motion with electrophysiology and high-throughput kinematic analysis of locomotor behaviors. Our results demonstrate that mechanosensory feedback enhances swimming speed by providing excitation to spinal circuits controlling the frequency of motor output during fast locomotion, thereby controlling the timing of spinal microcircuit activation and the transition between fast and slow locomotion. As it allows zebrafish larvae to travel the same distance in a shorter time (about 15 ms), mechanosensory feedback also bears ecological implications for fitness. By speeding up escapes during a predator attack, mechanosensory feedback may help enhancing survival.

To probe the contribution of mechanosensory feedback during active locomotion in zebrafish, we used the bioluminescent GFP-Aequorin sensor in motor neurons. We found that the amplitude of Aequorin signals from motor neurons could be used as a signature for distinguishing locomotor maneuvers including slow swimming and fast escape responses that have specific ranges of swimming frequencies and distinct profiles of motor neurons recruitment (*Liao and Fetcho, 2008*; *Masino and Fetcho, 2005*; *McLean et al., 2008*). We found that eliminating mechanical sensory

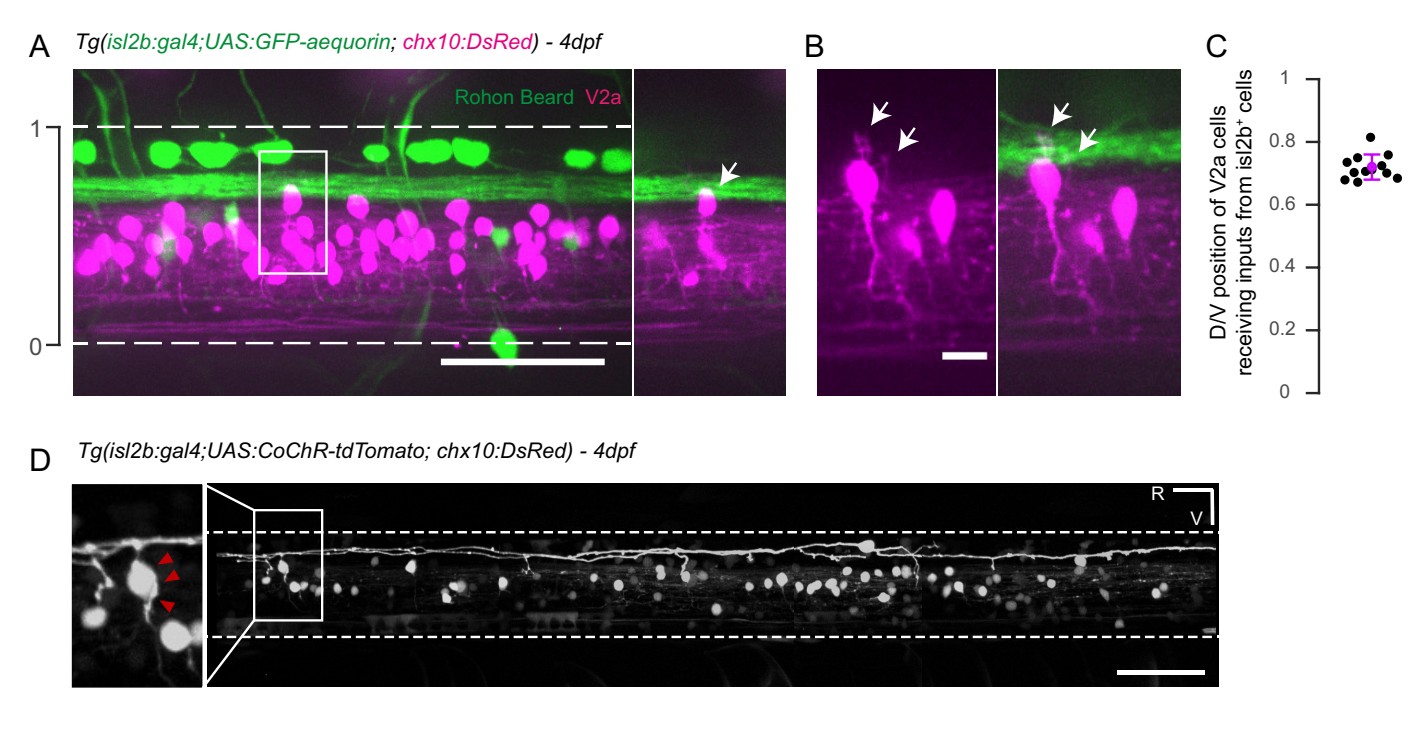

**Figure 6.** Axons of *isl2b*[+] Rohon-Beard project onto dorsalmost *chx10*[+] V2a interneurons in the spinal cord. (**A**) Z-projection stack of the spinal cord imaged from the lateral side in a 4 dpf *Tg(isl2b:gal4,cmcl2:eGFP;UAS:GFP-aequorin;chx10:DsRed)* triple transgenic larva. White dashed lines delineate ventral and dorsal limits of the spinal cord; these limits define the dorso-ventral (D/V) axis from 0 to 1. The axon bundle from *isl2b*[+] Dorsal Root Ganglia (DRG) and Rohon-Beard (RB) neurons contact the soma of dorsal *chx10*[+] V2a interneurons. The white square is enlarged in the right panel to stress the anatomical connections (arrow). (**B**) Dendrites of dorsal V2a interneurons are targeted by axonal processes from *isl2b*[+] sensory neurons (arrows). (**C**) D/V positions of V2a neurons receiving putative inputs from *isl2b*[+] cells (mean D/V position = 0.72 ± 0.04. N = 12 cells in n = 4 larvae). (**D**) Profile of expression of 4 dpf *Tg(isl2b:gal4,cmcl2:eGFP;chx10:DsRed)* injected with the *Tg(UAS:CoChR-tdTomato)* construct reveals projections from single Rohon Beard cells onto *chx10*[+] dorsal V2a interneurons. Scale bars in (**A**) and (**D**) are 50 μm and is 10 μm in (**B**). For each panel rostral side (R) is on the left and ventral side (V) is at the bottom.

feedback originating from muscle contraction led to a decrease of bioluminescence signals from motor neurons during acoustic escape responses. Our observations suggest that mechanosensory feedback enhances the recruitment of motor circuits. However, the effect observed could be driven by a decrease in swimming frequency and /or a decrease in the number of active spinal motor neurons. The poor temporal resolution of GFP-Aequorin prevented us from resolving these possibilities.

## Speed modulation by mechanosensory feedback

To determine how the deprivation of mechanosensory feedback affects motor neuron recruitment, we analyzed kinematic parameters of animals deprived of excitatory synaptic transmission in gluta-matergic mechanosensory neurons using the genetically encoded Botulinum toxin. The perturbation of major kinematic parameters such as left/right coordination or locomotor frequency during fictive locomotion usually requires the ablation of entire classes of spinal neurons (*Crone et al., 2008*; *Gosgnach et al., 2006*; *Talpalar et al., 2013*; *Zhang et al., 2008*). To identify the consequence of the selective silencing of mechanosensory feedback in the zebrafish spinal cord, we implemented a high-throughput kinematic analysis of acoustic escape responses in larvae deprived of neurotransmission in *isl2b*[+] mechanosensory neurons versus their control siblings. Our analysis revealed that mechanosensory feedback increases the swimming frequency and corresponding locomotor speed without affecting the total number of oscillations during an escape. Inputs from mechanosensory neurons first contribute to reaching the maximum range of fast swimming frequencies and second set the transition time from fast to slow swimming frequencies. Without mechanosensory inputs, the

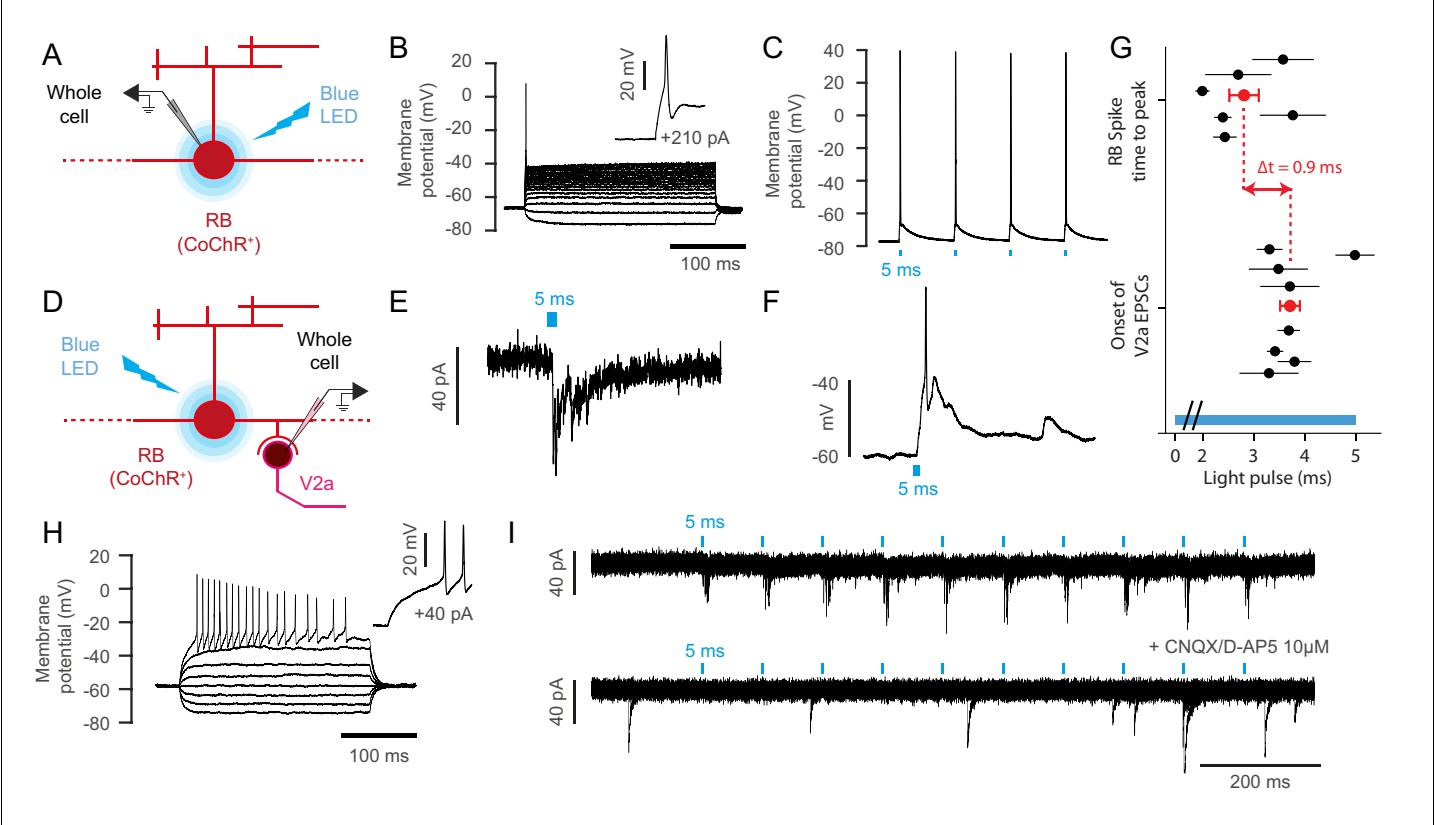

**Figure 7.** Rohon-Beard neurons selectively synapse onto ipsilateral dorsalmost V2a interneurons. (**A**) Schematic of the optogenetic calibration of CoChR-mediated stimulation in RB neurons. (**B**) Current-injection steps recording reveals the low-input resistance nature of Rohon Beard neurons, n = 7 cells). (**C**) 5 ms blue light pulse stimulations reliably elicit single action potentials in CoChR-expressing RB cells (n = 3 cells). (**D**) Schematic of whole cell recording of V2a interneurons with optogenetic stimulation of RB neurons. (**E**) Voltage clamp recording of a target V2a interneuron receiving monosynaptic inputs from a RB neuron. (**F**) Current clamp recording of a target V2a interneuron firing an action potential following the stimulation of a RB neuron. (**G**) Timing of RB spike time-to-peak (2.8 ms ±0.3 ms after the onset of the blue light pulse, n = 103 stimulations recorded in 6 cells) and onset of EPSCs in target V2a interneurons (3.7 ms ±0.2 ms after the onset of the blue light pulse, n = 78 EPSCs in 8 cells). The average lag between RB spiking and V2a EPSC is 0.9 ms. The blue bar represents the duration of the blue light pulse. (**H**) Membrane properties of recorded V2a interneurons in the dorsal cord, n = 5 cells). (**I**) Blockade of glutamatergic neurotransmission following the application of 10 μM of CNQX and D-AP5 suppresses elicited post-synaptic currents in a V2a interneuron (n = 2 cells).

initial swimming frequency is reduced and the transition from fast to slow swimming occurs earlier, which leads to an increase of locomotor event duration for the same number of oscillations and distance travelled, and therefore an overall slower escape response.

Contrary to mammals, there are no muscle spindles relaying mechanical information on muscle contraction to spinal circuits in fish. Instead, glutamatergic stretch receptor neurons that can sense body elongation in lamprey (*Grillner et al., 1981*; *Rovainen, 1974*) project onto spinal sensory neurons, motor neurons and interneurons (*Di Prisco et al., 1990*) and can entrain the locomotor central pattern generators (CPGs, [*Grillner et al., 1981*; *McClellan and Sigvardt, 1988*]). Recently, GABAergic cerebrospinal-fluid contacting neurons have been shown to detect body curvature following muscle contraction in both zebrafish (*Böhm et al., 2016*; *Grillner et al., 1984*; *Jalalvand et al., 2016*). Interestingly, the activation of each of these pathways, either during natural mechanical oscillations of the tail or optogenetic stimulations, is sufficient to trigger locomotor activity demonstrating that mechanosensory feedback can provide inputs to locomotor central pattern generator circuits in the spinal cord (*Böhm et al., 2016*; *Fidelin et al., 2015*; *McClellan and Grillner, 1983*; *Wyart et al., 2009*). Yet, the molecular identity of CPG neurons receiving inputs from mechanosensory neurons has remained unclear.

## Circuit architecture of mechanosensory neurons in zebrafish

Previous studies have described a remarkably simple and conserved disynaptic cutaneous skin reflex relying on spinal mechanosensory neurons referred to as Rohon Beard (RB) neurons in *Xenopus* and zebrafish and 'dorsal cells' in lamprey (*Buchanan and Cohen, 1982*; *Christenson et al., 1988*; *Clarke et al., 1984*; *Easley-Neal et al., 2013*; *Li et al., 2003*; *Roberts et al., 1983*). In these species, spinal mechanosensory neurons respond to skin stimulation at rest by firing a single spike and project onto glutamatergic commissural interneurons (referred to as dlc, CoPA and giant interneurons respectively [*Buchanan and Cohen, 1982*; *Easley-Neal et al., 2013*; *Li et al., 2003*; *Pietri et al., 2009*; *Sillar and Roberts, 1988*]), which drive contralateral motor neuron activation and subsequent swimming away from the stimulus (*Buchanan and Cohen, 1982*; *Christenson et al., 1988*; *Li et al., 2003*, *2004b*). During swimming, excitation of commissural sensory interneurons from sensory neurons was shown to be filtered out by phase-locked inhibition (*Knogler and Drapeau, 2014*; *Sillar and Roberts, 1988*), ensuring that the recruitment of RB neurons potentially arising from movement-related activation would not trigger the disynaptic cutaneous reflex.

Here we discovered a novel sensory disynaptic pathway originating from spinal sensory neurons. We demonstrate that mechanosensory Rohon-Beard neurons synapse onto dorsalmost V2a interneurons previously shown to drive fast swimming in zebrafish (*Ampatzis et al., 2014*; *McLean et al., 2008*, *2007*). Furthermore, we show that in some cases, this monosynaptic connection can provide suprathreshold inputs to V2a interneurons recruited during fast locomotion. Given that the *isl2b* transgenic line used in this study also labels dorsal root ganglia and trigeminal neurons, it is possible that the modulation of speed results from synergistic inputs from these three neuronal populations (*Buhl et al., 2012*). Dorsal root ganglia might in particular provide glutamatergic sensory inputs that could modulate locomotor frequency as well. Yet, we identified here a direct RB-V2a pathway that could contribute to the enhancement of speed triggered by mechanosensory feedback during the escape. The contribution of RB neurons to the enhancement of speed by mechanosensory feedback is consistent with previous experiments showing that stimulation of RB neurons can reset or increase locomotor frequency in *Xenopus* tadoples (for example,[*Sillar and Roberts, 1992*]). The role of mechanosensory feedback to increase swimming speed is consistent with former observations in lamprey showing an entrainment of locomotion by mechanical stimulation of the spinal cord (*Grillner et al., 1981*). Therefore, the direct connection we demonstrated here from spinal mechanosensory neurons onto a subtype of ipsilateral V2a interneurons recruited during fast swimming could underlie the effect we observed on swimming speed, along with other potential pathways involving dorsal root ganglia and trigeminal neurons.

RB neurons have complex processes branching under the skin as well as ascending and descending in the spinal cord. One critical experiment in the future will consist of recording the activity of RB neurons during muscle contraction in freely swimming animals. Recording reliable calcium transients associated with mechanical stimulation has been possible by combining fluorescent calcium sensors in one wavelength with another fluorescent protein enabling to control for cell position (*Böhm et al., 2016*). Nonetheless measuring activation of RB neurons during mechanical stimulation remains challenging due to the fact that RB neurons usually fire single action potentials ( *et al., 2017*; *Roberts et al., 2012*; *Winlove and Roberts, 2012*) and most calcium indicator fail to report single spike without extensive averaging (see *Chen et al. [2013]*; *Tian et al. [2009]*). Further studies will identify the connectivity diagrams of each of these mechanosensory cell types and devise genetic strategies to silence these neurons individually to probe their contribution to speed modulation during active locomotion. While local reflex pathways have been characterized at rest, further studies will decipher the mechanisms underlying the diversity of behavioral effects due to mechanosensory neuron recruitment: escapes for single touch at rest (*Buchanan and Cohen, 1982*; *Douglass et al., 2008*; *Roberts et al., 1983*), struggling for continuous stimulation (*Li et al., 2007b*) and enhancement of speed during active locomotion (this study).

## Conclusion

Altogether, our results about the role of mechanosensory feedback in speed modulation can explain the general observation made in various invertebrate as well as vertebrate species that locomotion is faster in active compared to fictive conditions. In invertebrates, there is evidence in several species for fictive locomotor patterns being typically twice slower than real locomotion (sea slugs,

(*Morton and Chiel, 1993*), 1993; crayfish, (*Bacqué-Cazenave et al., 2015*); and fruit fly, (*Mendes et al., 2013*). The leech is a particularly interesting example as it can perform both swimming and crawling. The effect of denervation is a systematic slowdown of locomotor activity, although it is much larger in crawling (twofolds) (*Baader and Kristan, 1992*; *Eisenhart et al., 2000*; *Kristan and Calabrese, 1976*). In lamprey where the same animal was used to compare active and fictive locomotion, the frequency of fictive swimming appears 5–10 times slower than the frequency of active swimming recorded through electromyography (*Wallén and Williams, 1984*). The general slowdown of locomotor frequencies when peripheral sensory feedback is removed is commonly observed in fictive locomotion in mice as well (*Bellardita and Kiehn, 2015*; *Talpalar et al., 2013*). New preparations to manipulate and record from spinal neurons in moving mice (*Hayes et al., 2009*; *Hochman et al., 2012*) will enable in the future to investigate the dynamic role of mechanosensory feedback during active locomotion in mammals.

Our study explains how mechanosensory neurons enhance speed during active locomotion and reveal one possible spinal circuit that could mediate this effect. Furthermore, our work demonstrates that microcircuit selection during locomotor behaviors is not only set by descending commands from supra-spinal centers but could rather result from a combination of inputs from descending and local spinal sensorimotor circuits recruited while the movement is performed.

## Material and methods

### Zebrafish care, generation and characterization of transgenic lines

All procedures were approved by the Institutional Ethics Committee at the Institut du Cerveau et de la Moelle épinière (ICM), Paris, France, the Ethical Committee Charles Darwin and received subsequent approval from the EEC (2010/63/EU). Adult AB and and Tüpfel long fin (TL) strains of *Danio rerio* were maintained and raised on a 14/10 hr light cycle and water was maintained at 28.5°C, conductivity at 500 µS and pH at 7.4. Embryos were raised in blue water (3 g of Instant Ocean salts and 2 mL of methylene blue at 1% in 10 L of osmosed water) at 28.5°C during the first 24 hr before screening for GFP expression. The *Tg(mnx1:gal4)^icm11* line was based on the injection of the *mnx1* construct provided by Dr. Thomas Auer and Dr. Filippo Del Bene (Institut Curie, Paris, France, see *Table 1*). The original sequence for GFP-Aequorin provided by Dr. Ludovic Tricoire (Université Pierre et Marie Curie, Paris, France) was subsequently codon-optimized for expression in zebrafish and subcloned into the PT2 14XUAS plasmid provided by Pr. Koichi Kawakami (National Institute of Genetics, Mishima, Japan). Injection of the *Tg(UAS:GFP-aequorin-opt)* construct in *Tg(mnx1:gal4)^icm23* larvae allowed the generation of the *Tg(mnx1:gal4;UAS:GFP-aequorin-opt)^icm09* line with selective expression of GFP-Aequorin in spinal motor neurons: more prominently primary dorsal motor neurons but also intermediate and ventral secondary motor neurons (*Figure 1B*) without any expression in the muscles and only very limited expression in the brain and hindbrain (note we cannot exclude that a minority a spinal ventral neurons distinct from motor neurons might be targeted). The transgenic line *Tg(UAS:GCaMP6f,cryaa:mCherry)^icm06* was generated by subcloning GCaMP6f (*Chen et al., 2013*) into pDONR221 and then assembled into the final expression vector in a three-fragment Gateway reaction using p5E-14XUAS, pME-GCaMP6f, p3E-poly(A) and pDest-CryAA:

**Table 1.** Stable transgenic lines used or generated in this study.

| Name | Original reference |
| --- | --- |
| *Tg(mnx1:gal4)^icm23* | (*Böhm et al., 2016*) |
| *Tg(isl2b:gal4,cmlc2:eGFP)* | (*Auer et al., 2015*) |
| *Tg(UAS:GFP-aequorin-opt)^icm09* | This paper |
| *Tg(UAS:GCaMP6f,cryaa:mCherry)^icm06* | (*Böhm et al., 2016*) |
| *Tg(chx10:loxP:DsRed:loxP:GFP)* | (*Kimura et al., 2006*) |
| *relaxed (cacnb1^ts25)* | (*Granato et al., 1996*) |
| *Tg(UAS:BoTxLCB-GFP)^icm21* | (*Auer et al., 2015*; *Böhm et al., 2016*; *Sternberg et al., 2016*) |

mCherry. Given the potential variability in expression patterns of Gal4 lines with different UAS, we characterized the *Tg(mnx1:gal4;UAS:GCaMP6f,cryaa:mCherry)* line and observed a similar expression in motor neurons as in the *Tg(mnx1:gal4;UAS:GFP-aequorin-opt)*[icm09] line. In both cases, we observed the targeting of spinal neurons exiting the spinal cord in the ventral root, and therefore referred to as motor neurons. However, we cannot exclude that the *mnx1* promoter may target other spinal neurons, distinct from motor neurons. *relaxed* mutants (*cacnb1*[ts25/ts25]) (*Granato et al., 1996*) were provided by Pr. Paul Brehm (Vollum Institute, Oregon Health and Science University, Portland, USA). In homozygous *cacnb1*[ts25/ts25] mutants, a mutation of the skeletal muscle dihydropyridine receptor *β*1a subunit interferes with the calcium release and mutant larvae are immotile (*Granato et al., 1996*). The *Tg(isl2b:gal4,cmlc2:eGFP)* line (*Auer et al., 2015*) driving expression in Dorsal Root Ganglia (DRG), trigeminal, Rohon-Beard neurons, a subset of Retinal Ganglion Cells (RGCs), and blood vessels was provided by Dr. Thomas Auer and Dr. Filippo Del Bene (Institut Curie, Paris, France) (see *Table 1*).

## Immunohistochemistry

Larvae were fixed in 4% PFA for 4 hr at 4°C followed by 3 × 5 min washes in PBS. Larvae were blocked for 1 hr in blocking solution (10% NGS, 1% DMSO, 0.5% Triton X100 in 0.1 M PBS). Larvae were incubated with the primary antibody over night at RT (1% NGS, 1% DMSO, 0.5% Triton X100 in 0.1% PBS). After washing three times for 5 min in 0.1 M PBST with 0.5% Triton X100 (PBST), larvae were incubated in the dark with the secondary antibody in PBST. After washing again three times for 5 min in PBST, larvae were mounted on a slide with mounting medium and were imaged on an upright confocal microscope (FV-1000, Olympus, Tokyo, Japan). The following antibodies were used: anti-GFP (Abcam ab 13970, dilution 1:500), Alexa Fluor 488 goat anti-chicken IgG (A11039, dilution1:1000, Invitrogen, Carlsbad, CA, USA). Immunostaining specificity was established by omitting the primary antibody.

## Monitoring neuronal activity with bioluminescence

Embryos were dechorionated and soaked at 26°C in 100 µL of blue water with a final concentration of 60 µM of coelenterazine-h (Biotium, Fremont, CA, USA). Coelenterazine-h was renewed at 2 dpf. All experiments were performed at 4 dpf. In all experiments, one larva was head-embedded in 1.5% low-melting point agarose with the tail free to move in a circular (2 cm diameter) 3D-printed arena (Sculpteo, Villejuif, France). The arena was then placed in a lightproof box (*Figure 1A*) and attached to a small speaker (2 Ohm). Each trial consisted of a 500 ms baseline followed by a 10 ms acoustic stimulus and 1990 ms subsequent recording. Assays consisted of 30 trials with 1 min inter trial intervals to reduce habituation. Sinusoidal stimuli (5 cycles, 500 Hz) were delivered through a wave generator (33210-A, Agilent, Santa Clara, CA, USA) and audio amplifier (LP2020A, Lepai). Intensity was adjusted to the lowest value that reliably elicited an escape response (between 0.5 and 5 Vpp). The same larvae used for the active assay were subsequently paralyzed by bath application of pancuronium bromide (P1918, Sigma-Aldrich, Saint-Louis, MO, USA) at 0.6 mg/mL final concentration and stimulation intensity was adjusted to the lowest value that elicited a bioluminescent signal. For *Figure 3*, *cacnb1*[ts25/ts25] and control siblings were tested alternatively on the same day and compared to each other. In non-moving animals (that is, paralyzed or *cacnb1* mutants), the intensity was progressively increased until stimuli elicited bioluminescence signals. A higher intensity of the acoustic stimulus was often needed after addition of pancuronium bromide, possibly due to modulation of cholinergic arousal brain circuitry (*Yokogawa et al., 2007*). As negative controls, bioluminescence assays of wild type animals or *Tg(mnx1:gal4;UAS:GFP)* (*Zelenchuk and Brusés, 2011*) where motor neurons express GFP only revealed no signals (n = 3 wild type larvae with 30 trials each, n = 5 *Tg (mnx1:gal4;UAS:GFP)* larvae with 30 trials each). Animals deprived of GFP-Aequorin did not produce any signals above baseline noise level during escape responses. Infrared light illumination for monitoring larval behavior was provided by an 850 nm LED (Effisharp, Effilux, France) mounted with 2 long-pass 780 and 810 filters (Asahi ZIL0780 and Asahi XIL0810, respectively) and a diffuser (DG10-120B, Thorlabs, Newton, NJ, USA). Video acquisition was performed at 1000 Hz using a high-speed infrared sensitive camera (Eosens MC1362, Mikrotron, Unterschleissheim, Germany; objective Nikkor 50 mm f/1.8D, Nikon, Tokyo, Japan) at 320 × 320 pixels resolution controlled by the software Hiris (RD Vision, Saint-Maur-des-Fossés, France). Photons were counted with a photomultiplier tube

(PMT, H7360-02, Hamamatsu, Japan) located under the larva arena and sent to an acquisition card (NI PCI 6602, National Instruments, Cos Cob, CT, USA). A band-pass filter (525 nm / 50 nm, ref. 489038–8002, Carl Zeiss, Thornwood, NY, USA) and a short-pass filter (670 nm, XVS0670, Asahi, Japan) were placed between the larva and the PMT. A custom application-programming interface developed in collaboration with R&D Vision synchronized the video acquisition with the photon count and the stimulus delivery using 30 trials batched TTL chronogram (EG Chrono, RD Vision, France).

## Kinematics and bioluminescence analysis

The base and tip of the tail were manually determined for each larva and the tail was subsequently automatically tracked with a custom Matlab algorithm (R2012b, Mathworks, Natick, MA, USA). The tail angle was computed for each frame and filtered using median filtering (window size = 10). The start of the movement was determined as the first frame followed by 3 with a differential tail angle value above 0.08. The end was determined as the end of the 20 frames with a differential tail angle value below 0.1 degree. Local minimal and maximal values of the tail angle occurred at least 2 ms apart and 1 degree above the 5 ms preceding value. Automated movement categorization was determined as follows: 'escapes' for all movements starting with an asymmetrical C-bend and number of cycles $\geq$ 1; 'slow swims' for all movements with symmetrical bends of amplitude <25° and number of cycles > 1.

Photons were counted with a temporal resolution of 1 ms and then binned every 10 ms. The signal was filtered using a running average with a window size of 10, giving a typical signal-to-noise ratio (SNR) for active movements of 50 to 1. Noise was extrapolated from a linear fit of the cumulative photon count before the stimulus and subtracted from the signal. The start and end of the bioluminescent signal were computed as the first time point followed by three points with a differential value above 0.4 photons/10 ms and below 0.2 photons/10 ms, respectively. The time-to-peak was calculated between the start and the peak of the bioluminescent signal while the decay coefficient was derived from the one-term exponential fit between the peak and the end of the signal.

## Calcium imaging of spinal motor neurons and ventral nerve root recordings (VNR)

4 dpf *Tg(mnx1:gal4; UAS:GCaMP6f,cryaa:mCherry)* double transgenic larvae were screened for dense labeling and good expression of GCaMP6f in spinal motor neurons under a dissecting microscope equipped with an epifluorescence lamp (Leica, Wetzlar, Germany). Larvae were anaesthetized in 0.02% Tricaine-Methiodide (MS-222, Sigma-Aldrich) diluted in fish facility water and mounted on their lateral side in 1.5% low-melting point agarose in glass-bottom dishes filled with external solution ([NaCl] = 134 mM, [KCl] = 2.9 mM, [MgCl2] = 1.2 mM, [HEPES] = 10 mM, [glucose] = 10 mM and [CaCl2] = 2.1 mM; adjusted to pH 7.7–7.8 with NaOH and osmolarity 290 mOsm). Larvae were immobilized by injecting 0.1–0.3 nL of 0.5 mM α-Bungarotoxin (Tocris, Bristol, UK) in the ventral axial musculature. A portion of agar was removed using a razor blade in order to expose 2 to 3 segments. To achieve a strong signal-to-noise ratio during fictive locomotion recordings, the skin overlying these segments was removed using suction glass pipettes. Zebrafish larvae were imaged using a custom spinning disk microscope (Intelligent Imaging Innovation, Denver, CO, USA) equipped with a set of water-immersion objectives (Zeiss 20X, 40X, NA = 1). Recordings were acquired using Slidebook software at 20 Hz using a 488 nm laser. Gain and binning were optimized to maximize signal to noise ratio. Z-projection stacks showed full pattern of expression using Fiji (*Schindelin et al., 2012*). Positions of cells along the D-V axis were computed using Fiji and Matlab (Mathworks, USA). Calcium signals were extracted online using custom scripts. Regions of interest (ROIs) were manually designed and calcium signals time series were extracted as the mean fluorescence from individual ROIs at each time point of the recording. We observed that out-of-focus signals varied between animals, from dorsal to ventral spinal cord regions in a behavior-dependent manner. To estimate the contribution of out-of-focus signals we systematically picked two background ROIs, one placed below the ventral limit of the spinal cord to capture out-of-focus signals at the level of ventral motor neurons during slow swimming, the second in the dorsalmost part of the spinal cord to capture out-of-focus signals in the dorsal spinal cord during the escape. We estimated the maximum out-of-focus signals observed during each behavior and used this value as a threshold for discriminating active

from silent motor neurons. Thin-walled, borosilicate glass capillaries (Sutter Instruments, Novato, CA, USA) were pulled and fire-polished from a Flaming/Brown pipette puller (Sutter Instruments, Novato) to obtain peripheral nerve recording micropipettes. Pipettes were filled with external solution and positioned next to the preparation using motorized micromanipulators under the microscope. Light suction was applied when the pipette reached the muscle region located at the vicinity of intermyotomal junctions, ventral to the axial musculature midline. VNR signals were acquired at 10 kHz in current clamp IC = 0 mode using a MultiClamp 700A amplifier (Molecular Devices–Axon Instruments, Sunnyvale, CA, USA), a Digidata series 1322A digitizer (Axon Instruments) and pClamp 8.2 software (Axon instruments). Recordings were considered for analysis when the background noise did not exceed 0.05 mV amplitude and signal to noise ratio for fictive locomotor events detection was above three. VNR recordings were analyzed offline and aligned to calcium imaging data using custom-made Matlab scripts.

## Fluorescence-guided whole-cell recordings and pharmacology

4 dpf larvae were pinned down in Sylgard-coated, glass-bottom dishes filled with external solution with thin tungsten pins through the notochord. Skin was removed from segments 5–6 to the end of the tail using sharp forceps. After removing the skin, one to two segments were dissected using glass suction pipettes. Patch pipettes (1B150F-4, WPI, Sarasota, FL, USA) were designed to reach a tip resistance of 8–12 MOhms and were filled with potassium-containing internal solution (concentrations in mM: K-gluconate 115, KCl 15, MgCl2 2, Mg-ATP 4, HEPES free acid 10, EGTA 0.5, 290 mOsm, adjusted to pH 7.2 with KOH and supplemented with Alexa 647 at 4 mM). Cells were held at around −60 mV. We calculated the liquid junction potential in our experiments (−9 mV) but did not correct for it since it did not affect the outcome of our experiments. NMDA receptor antagonist (D-AP5, Tocris) and AMPA receptor antagonist (CNQX, Tocris) were bath applied at 10 µM final concentrations. Kinetic parameters of light-evoked currents and EPSCs were extracted and analyzed using custom-made Matlab scripts.

## Behavioral analysis of freely moving BoTxBLC larvae

*Tg(isl2b:gal4,cmlc2:eGFP;UAS:BoTxBLC-GFP)* larvae were screened at 2 dpf for expression. At 6 dpf, larvae were tested 4 by 4: each larva was positioned in a separate dish (2 cm diameter) and illuminated from below, freely moving. Each swim arena contained 400 µL of external solution to minimize movement in the z-axis. Escapes were elicited by delivering a 500 Hz stimulus for 1 ms via 20W speakers. Each trial consisted of a 200 ms pretrial window followed by the stimulus and 800 ms subsequent recording. Five trials were performed in succession with 2 min long intertrial resting periods. Behavior was recorded at 650 fps with a high-speed camera (acA2000-340km, Basler, Ahrensburg, Germany) and analyzed using a tracking algorithm (ZebraZoom, [*Mirat et al., 2013*]) and a custom Matlab script. Extracted kinematic variables are listed and defined as follows: duration (the amount of time during movement), distance (travelled distance), speed (distance/duration), number of oscillations (1/2 of the number of peaks in tail angle over time), and mean tail beat frequency (TBF, inverse of the duration between two subsequent peaks in the tail angle). Visual inspection of the unsmoothed tail angle was used to exclude trials in which erroneous tracking occurred. Trials fulfilling at least one of the following criteria were programmatically excluded: (i) Detected movement prior to the stimulus. In these cases, spontaneous pretrigger movement lingering into the post-trigger recording window can occlude escape-type movements. (ii) First peak in tail-angle occurring more than 20 frames after the stimulus. These no longer represent low-latency escapes and most likely represent a different category of movement altogether (*Budick and O'Malley, 2000*; *Liu and Fetcho, 1999*). (iii) First peak in tail-angle lower than 60 degrees. Bouts lacking an initial C-bend are not escapes and are most likely slow swim bouts or other movement types (*Budick and O'Malley, 2000*). Hemi-periods were calculated as the interval between two consecutive peaks in the tail angle, and subsequently used as a determinate to extract fast and slow components of the escape with a cutoff of 650/60 frames (30 Hz). Peaks which alternated between the cutoff value after the first appearance of a slow peak were excluded.

## Statistical analysis

SPSS 20 (IBM, Armonk, NY, USA) was used to perform all statistical analyses. Comparisons of bioluminescence signals parameters was conducted using a t-test for paired samples for repeated measures within subjects (that is, active versus paralyzed data). Mixed linear model analysis with repeated measures using an auto-regressive covariance structure was performed to compare the bioluminescence amplitude between movement categories in active assays, and between moving and immotile larvae in active versus fictive assays. The same model was used to analyze behavioral parameters in order to account for repeated measures in within-fish (trial number) and between-group comparisons (genotype). Bioluminescence time decay coefficients were included if the goodness of fit r-square value was >0.95. A Pearson test was used to assess correlations for parametric data. Statistical significance is represented in the graphs as *** for p<0.001, ** for p<0.01, * for p<0.05, corrected for multiple comparisons when needed. All data are provided in the figures and text as means +/- standard error of the mean (SEM).

## Acknowledgements

We would like to thank Dr. Ludovic Tricoire (University Pierre et Marie Curie, France) for providing the original GFP-Aequorin sequence, Dr. Michael Granato (University of Pennsylvania, USA), Dr. Paul Brehm (Vollum Institute, USA) for providing the *relaxed (cacnb1^{ts25})* mutants, Dr. Maximilliano Suster and Prof. Koichi Kawakami (NIG, Japan) for sharing the *Tg(UAS:BoTxBLC-GFP)* transgenic line, Dr. Tom Auer and Dr. Filippo Del Bene (Institut Curie, France) for providing the *mnx1* and *isl2b* constructs and the *Tg(isl2b:gal4,cmlc2:eGFP)* line, and Dr. Minoru Koyama (Janelia Farm, USA) for the *UAS:CoChR-tdTomato* construct. We thank Dr. Andrew Straw (University of Freiburg, Germany), Dr. Björn Brembs (University of Regensburg, Germany), and Dr. William Kristan (UCSD, USA) for their valuable feedback. We are indebted to RD Vision for developing the custom API to synchronize photon collection and video acquisition. We thank Bogdan Buzurin, Monica Dicu and Natalia Maties for fish care. SK received a PhD fellowship from Inserm and Assistance Publique – Hôpitaux de Paris, KF from University Pierre et Marie Curie, and UB from Ecole des Neurosciences de Paris (ENP). AP received a postdoctoral fellowship from the Mairie de Paris Research in Paris program and from the EMBO. This work received financial support from the Institut du Cerveau et de la Moelle épinière with the French program 'Investissements d'avenir' ANR-10-IAIHU-06, the ENP Chair d'excellence, the Fondation Bettencourt Schueller (FBS), the City of Paris Emergence program, the ATIP/Avenir junior program from INSERM and CNRS, the NIH Brain Initiative grant no. 5U01NS090501 and the European Research Council (ERC) starter grant 'OptoLoco' #311673.

## Additional information

### Funding

| Funder | Grant reference number | Author |
| --- | --- | --- |
| European Research Council | 311673 | Claire Wyart |
| National Institutes of Health | 5U01NS090501 | Claire Wyart |

The funders had no role in study design, data collection and interpretation, or the decision to submit the work for publication.

### Author contributions

SK, APr, Data curation, Formal analysis, Investigation, Writing—original draft, Writing—review and editing; KF, Data curation, Formal analysis, Investigation, Writing—original draft, Writing—review and editing ; P-EBT, Data curation, Formal analysis, Investigation ; APa, Data curation, Formal analysis ; CD, Data curation, Formal analysis; ULB, SNF, OT, Data curation ; HP-M, Conceptualization ; CW, Conceptualization, Resources, Formal analysis, Supervision, Funding acquisition, Validation, Investigation, Visualization, Methodology, Project administration, Writing—original draft, Writing—review and editing

## Author ORCIDs
Steven Knafo, http://orcid.org/0000-0002-8733-0429
Kevin Fidelin, http://orcid.org/0000-0001-7676-1410
Olivier Thouvenin, http://orcid.org/0000-0003-4853-7555
Claire Wyart, http://orcid.org/0000-0002-1668-4975

## Ethics

Animal experimentation: All procedures were approved by the Institutional Ethics Committee at the Institut du Cerveau et de la Moelle épinière (ICM), Paris, France, the Ethical Committee Charles Darwin and received subsequent approval from the EEC (2010/63/EU).

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
