## [Decision Letter]

Thank you for submitting your article "Mechanosensory Neurons Control the Timing of Spinal Microcircuit Selection during Locomotion" for consideration by *eLife*. Your article has been reviewed by three peer reviewers, one of whom, Ronald L Calabrese (Reviewer #1), is a member of our Board or Reviewing Editors and the evaluation has been overseen by Didier Stainier as the Senior Editor. The following individuals involved in review of your submission have agreed to reveal their identity: Mark A Masino (Reviewer #2); David McLean (Reviewer #3).

The reviewers have discussed the reviews with one another and the Reviewing Editor has drafted this decision to help you prepare a revised submission.

Summary:

This is an elegant exploration of how sensory feedback affects escape and slow swimming in zebra fish larvae. Combing genetic, imaging, electrophysiological, optogenetic and kinematic analysis, the authors show that mechanosensory feedback enhances the recruitment of motor pools and controls the speed and duration of escape swimming. They then show that Rohon Beard neurons (spinal cord mechanoreceptors) make anatomical contacts and functional synaptic connections with V2a interneurons that are known to control speed of escape swimming. The experiments are carefully done with many thoughtful controls. The data shown is sufficient and analyzed with appropriate statistics. This work reveals microcircuits that underlie speed modulation by mechanosensory feedback in the vertebrate spinal cord and will thus be of wide interest.

Essential revisions:

There is no direct causal evidence that Rohon Beard neurons do indeed mediate the speed changes associated with active swimming, because they are not selectively silenced. Moreover, there is no direct demonstration that they are activated by the animals own movements during active swimming. These weaknesses can be mitigated in two ways.

1) An experiment where calcium imaging is used to monitor transients in RB neurons in partially embedded versus fully embedded and/or paralytic fish. This would demonstrate whether the RB cells participate in reafference.

2) More extensive analysis of the connectivity data between RB neurons and V2a neurons, including increases in the number of independent observations. Latency repeatability and jitter should be critically evaluated both for the EPSC and for the spike response of V2a neurons to RB cell photoactivation. We would like to be reassured that the nature of the connection is truly distinct from the one revealed by prior work, and is thus capable of influencing V2a activity during swimming.

The detailed reviews of two of the reviewers are included to help in revision but the focus of revision should be the two major points above.

Reviewer #1

1) The paper is well written within text but important details are missing from the legends that allow easy interpretation of the data. Some figures are too dense as if composed for a journal with restrictions on figure numbers.

All legends must be upgraded so that a reader can easily understand the data. Let me focus on Figure 1 to illustrate what must be done. What is going on in panels A and G? What are the superimposed lines in panel C (I know this but I am just saying be complete)? What is the color code in panel H and how does it relate to panel I. In panel I, I am really confused please explain the traces. Why are there multiple bumps in the left traces and what does the superposition mean in each set of traces? Please write figure legends that allow the data to be easily understood. Figure 3 is also a problem. What do the arrow and the arrowhead point out in panel A? What are we looking at in the left most image of panel C and in the middle image of panel C and left image of panel D? What are the different colors showing? In all graphs like panel E1 the mean and SEM are just too difficult to see especially amid the blue points; make bold.

Split Panels G to K of Figure 1 into a separate figure. Consider splitting Figure 5 also.

2) I am concerned about the morphological data shown in Figure 1. Was this observation made in a number of larvae? In Figure 5, was this observation made in a number of larvae. The legend of panel C implies that it was but you must be explicit. In Figure 5, was this observation made in a number of larvae?

3) There is no direct causal evidence that Rohon Beard cells do indeed mediate the speed changes associated with active swimming, because they are not selectively silenced or optogenetically activated. This needs to be more explicitly acknowledged.

4) Given the central effects of pancuronium bromide why wasn't α-bungarotoxin used to immobilize the larvae in the experiments of Figure 2? I am also concerned that you did not control for stimulus intensity in these experiments. In active fish and during pancuronium paralysis, is the size of the aequorin response dependent of stimulus amplitude? If it is, then comparisons of amplitudes across these two groups is suspect because different stimulus amplitudes were used.

Reviewer #3:

1) If RB neurons are encoding proprioceptive information in addition to the cutaneous exteroceptive information they are normally associated with, then activation of RB neurons during acoustic stimuli needs to be confirmed. The prediction is that RB neurons would respond to skin stimulation and acoustic stimulation when the tail can move, but not to acoustic stimuli when the tail is immotile. Without a better idea of whether RB neurons are in fact responding to self generated movements, then it is difficult to attribute the behavioral deficit to them and justify the ensuing circuit busting. For example, it could be that there are neurons in the DRG that are instead responsible.

2) There has been quite a bit of work in *Xenopus* tadpoles studying RB circuitry involved in cutaneous skin reflexes (e.g., Li et al., 2003, J Neurosci.; Li et al., 2004, J Neurophysiol). Abrupt stimulation of RB afferents generates a reflexive response away from the stimulus. Also, as I understand it, they demonstrate that drive from RB cells that potentially arises from movement-related stimulation would largely be filtered out by inhibition (Sillar and Roberts, 1988, Nature). However, RB neurons can reset or increase swimming rhythms when they are stimulated (e.g., Sillar and Roberts, 1992, J Neurosci), so there is clearly the potential for RB neurons to act as they are proposed to here. I think it would be worth explaining more how the findings here fit into previous work in tadpoles, as well as studies of dorsal cells (the RB equivalents) in lampreys (Buchanan and Cohen, 1982, J Neurophysiol; Christenson etal.,1988, Brain Res). I think the role of the RB-V2a connection would be worth considering in the context of struggling as well (Li et al., 2007, J Neurosci).

3) The optogenetic-electrophysiology data as currently presented don't provide sufficient temporal resolution to determine if the RB-V2a connection is monosynaptic or not. If it is, this would be novel, since studies of RB circuits in *Xenopus* have only revealed indirect connections from RBs to V2a-like neurons (with extremely short latencies though). In fact, the responses have multiple components which would be more consistent with a polysynaptic connection (especially since RB neurons tend to fire once). Given this is not the main goal of the work, I would suggest perhaps stepping back from saying it is a direct connection and instead focus on the excitatory nature of that connection, monosynaptic or not. Of course, this could all be moot if the RBs are not even active during body bending.

4) For the botulinum experiments it is largely assumed that the sensory populations are silenced, but it would be better if there was some confirmation in this line of fish. Patch recordings would be ideal, but if not possible a mention of whether the fish fail to respond to touch or not would suffice.

5) Since the RB driven response is specific to high frequency swimming commensurate with large deflections of the tail, I was expecting that the controls for the aequorin experiments would be zebrafish that are fully embedded so the tail can't move. Is it known if there is still less motor neuron recruitment in this condition comparable to toxin and mutants?

6) Work in lampreys studying the role of edge-cells seems appropriate to discuss more, since the function appears to be very similar to that proposed here and a lot of that work is performed while the tail is moving. Also the recent development of a rodent preparation is another precedence for neural recordings in moving animals (Hayes etal., 2009, J Neurophysiol; Hochman et al., 2012, Front. Biosci).

[Editors' note: further revisions were requested prior to acceptance, as described below.]

Thank you for resubmitting your work entitled "Mechanosensory Neurons Control the Timing of Spinal Microcircuit Selection during Locomotion" for further consideration at *eLife*. Your revised article has been favorably evaluated by Didier Stainier (Senior editor), a Reviewing editor, and two reviewers.

The manuscript has been greatly improved but there are some remaining issues that need to be addressed before acceptance, as outlined below:

A few more changes are necessary that do not require re-review by the expert reviewers.

1) There was concern that the authors overstated the case for monosynapticity in the Introduction and Results, subsection “Mechanosensory neurons synapse onto V2a interneurons involved in fast locomotion***”***, given that experiments with a high concentration of divalent ions – or the like – to eliminate (reduce the probability of) disynaptic events were not performed. We suggest that the text starting in subsection “Mechanosensory neurons synapse onto V2a interneurons involved in fast locomotion***”*** be modified something like "To assess whether RB-V2A ipsilateral connection might be monosynaptic, we analyzed the lag between RB spikes time-to-peak and V2a EPSCs onset and found that it was below 1 ms (Figure 7, 0.9 ms), a value compatible with monosynaptic connection (see (Li et al., 2007a) where cutoff is set at 3 ms for monosynaptic connections)." The Introduction can be correspondingly edited.

2) Quoting one of the expert reviewers "There is an underlying assumption that all UAS lines (GFP-aequorin-opt and GCaMP6f;cryaa:mCherry) will produce the same expression patterns when paired with this Gal4 (mnx1:gal4) line. Just because the icm23 line (mnx1:gal4-UAS:preEGFP) was characterized by Bohm et al., this does not mean that Gal4 will work the same with different versions of UAS (UAS:GFP-aequorin-opt or GCaMP6f;cryaa:mCherry). Have these lines been characterized? If so, references to that work must be made to show MN specificity/variegated expression patterns (if any)."

Please make an effort to give any references that are relevant and provide any characterizations performed at least in Material and methods.

---

## [Author Response]

*Essential revisions:*

*There is no direct causal evidence that Rohon Beard neurons do indeed mediate the speed changes associated with active swimming, because they are not selectively silenced. Moreover, there is no direct demonstration that they are activated by the animals own movements during active swimming.*

We agree with the reviewers that an ideal strategy would consist in selectively targeting and silencing Rohon-Beard neurons during active locomotion. Yet, we searched extensively the literature and existing databases and could not find any transgenic line targeting RBs without trigeminal neurons & dorsal root ganglia (see Kucenas et al.,2006 Neuroscience 138:641–652; Scott et al.,2007 Nature Methods 4:323 as well as Koichi Kawakami’s and Harry Burgess’s enhancer trap libraries).

*These weaknesses can be mitigated in two ways.*

*1) An experiment where calcium imaging is used to monitor transients in RB neurons in partially embedded versus fully embedded and/or paralytic fish. This would demonstrate whether the RB cells participate in reafference.*

We thank the reviewers for suggesting these experiments. The technical challenge of reliably detecting calcium transients in RB neurons lies in the fact that these cells usually fire single spikes in response to current steps in zebrafish larva (see Figure 7 for one example and Figure 8). This is also the case in the tadpole, as previously shown by the group of Alan Roberts using current steps and gentle touch (Winlove and Roberts, 2012, EJN 36:2926-40). In particular, Alan Roberts and collaborators found that touch led to few spikes while stronger potentially damaging stimuli led to multiple spikes (Roberts et al.,2011 Dev. Neurobiol. 575-584).

Nonetheless, to assess the recruitment of RB neurons during swimming, we performed three independent sets of experiments.

1) First, we drove expression of genetically encoded calcium indicators GCaMPs in *isl2b+* neurons in order to monitor the recruitment of RB neurons during active escape responses.

With GCaMP6f (Kd ~ 250 nM in solution, Chen et al., Nature 2013 and Loren Looger *personal communication*), we did not reach enough expression in RB neurons to perform calcium imaging experiments. On the contrary, with GCaMP5G (Kd = 373nM in solution, Tian et al., Nature Methods 2009 and Loren Looger *personal communication*), expression of the sensor was satisfactory (see Figure 9). Yet, we observed calcium transients during escape responses only in a minority (2 out of 10) of fish expressing GCaMP5G in RB neurons (see examples from one responding fish depicted in Figure 9). As such, RB responses were not representative of the 10 tested fish so we did not include them in the main manuscript.

Author response image 1.Responses to RB neurons to electrical and optogenetic stimulations consists in single spikes (n = 5).RB neurons respond to a long current step by a single action potential (top row). RB neurons in *Tg(isl2b:gal4)* expressing the construct *UAS:CoChR-tdTomato* respond to 5ms light pulses by single action potentials (bottom row).**DOI:**
http://dx.doi.org/10.7554/eLife.25260.017

Author response image 2.Calcium transients recorded in RB neurons from a non representative zebrafish larva in *Tg(isl2b:gal4;cry:GFP,UAS: mRFP; UAS:GCaMP5G)* triple transgenic animals.(**A**,**B**) Expression of mRFP (**A**) and GCaMP5G (**B**) is restricted to RB neurons in this field of view. Successive snapshots show signals before the stimulus (0s), during tail motion (3 s), immediately after the motion artifact when the cells are back in the plane (3.5 s) and long after tail motion (9 s). (**C**) Calcium transients estimated as relative △F/F = △F (green) – △F (red) is depicted for four trials before (top) and after (bottom) application of the paralytic agent Pancuronium Bromide (PB) in the bath.**DOI:**
http://dx.doi.org/10.7554/eLife.25260.018

One issue is that even the ultra-sensitive genetically encoded calcium sensors GCaMP with Kd ~ 300nM are not sensitive enough to detect single spikes in vivowithout intensive averaging across trials and across neurons (check for example the averaging performed in Tian et al., 2009 Nature Methods 6:875-81 for GCaMP5F and in Chen, Wardill et al.,2013 Nature 499: 295- 300 for GCaMP6f). Therefore by using these sensors in vivo, we can easily miss calcium transients associated with single spikes in RB neurons in response to tail contractions.

2) In order to optimize the imaging of calcium in RB neurons, we therefore retrogradely labeled RB neurons by injections in the spinal cord of Calcium Green Dextrans (Kd ~ 190-270 nM in solution, see Paredes et al., 2008 Methods 46:143-151). We attempted to detect single spikes in 6 RB neurons using an assay where the spinal cord is mechanically stimulated with a glass probe (see Böhm et al.,Nature Communications 2016). Skin stimulations are known to reliably activate RB neurons, which in turn fire single action potentials. However, the Calcium Green Dextran approach led to variation of fluorescence associated with RB neuron firing in a minority of neurons (1 out of 6 tested RB neurons using the mechanical probe).

3) As fluorescent calcium sensors failed to show transients in single cells, we took advantage of the bioluminescence assay to monitor the activity of *isl2b^+^*neurons during acoustic escape responses before and after blockade of muscle contraction. GFP-Aequorin bioluminescence signals were detected during active tail contraction (see Figure 10) and were completely abolished during escape responses when muscles were paralyzed (see Figure 10). These data therefore suggest that *isl2b^+^*neurons are recruited during active locomotion. Yet, since the *isl2b* promoter targets multiple cell types (RB neurons, trigeminal, DRGs, along with blood vessels and retinal ganglion cells), this approach cannot prove that RB neurons are the main neurons recruited during active swimming.

Conclusion

Despite our efforts, the three approaches used to test whether RB neurons were recruited during active motion are not conclusive. This could be due to the difficulty of monitoring calcium transients associated with single spikes in moving animals. Consequently, we edited the manuscript:

a) to emphasize that behavioral effects observed using BoTxLC-GFP could be due to silencing of either RB neurons, trigeminal neurons or dorsal root ganglia (Abstract and Discussion);

b) to mention that we could not measure calcium transients in response to bending in RB neurons reliably (see Discussion).

Author response image 3.Probing the recruitment of Rohon-Beard neurons expressing GFP-Aequorin during movement.(**A**, **B**) Lateral view showing the expression of GFP-Aequorin in 4 dpf *Tg(isl2b:gal4;UAS:GFP- Aequorin-opt)* larvae throughout the animal (**A**) and in the spinal cord (**B**). The arrowhead points to one Rohon-Beard cell in the dorsal spinal cord. The white dashed lines delineate the ventral and dorsal limits of the spinal cord. As previously stated for *Tg(mnx1:gal4; UAS:GFP-Aequorin-opt)* larvae, we verified on 5 animals that expression never targeted muscle fibers in these double transgenic animals. Ventral is bottom, rostral is left. (**C**) Bioluminescence signals emitted from 10 *isl2b^+^*following an acoustic stimulus during motion (blue) or after blockade of muscle contraction using pancuronium bromide in the bath (red). (**D**) Quantification of the amplitude of bioluminescence signals across fish in active and paralyzed larvae (4.53 +/- 0.21 versus 2.20 +/- 0.10 photons / 10 ms, n = 10 fish, n = 600 trials, p < 0.001).**DOI:**
http://dx.doi.org/10.7554/eLife.25260.019

*2) More extensive analysis of the connectivity data between RB neurons and V2a neurons, including increases in the number of independent observations. Latency repeatability and jitter should be critically evaluated both for the EPSC and for the spike response of V2a neurons to RB cell photoactivation.*

We thank reviewers for pointing at this critical aspect. We analyzed the connectivity of 4 additional RB-V2a pairs, reaching a total of 8 out of 8 V2a interneurons in 8 different larvae receiving synaptic currents from ipsilateral RB neurons.

To demonstrate the monosynaptic nature of the RB-V2a connection, we first analyzed the time- to-peak of light-evoked action potentials in CoChR^+^ RB neurons (5-ms blue light stimulations, n = 103 stimulations recorded in 6 cells) as well as the delay of ESPCs in *chx10^+^*V2a neurons following RB stimulations (78 EPSCs recorded in 8 cells). Our analysis shows that RB fire single action potentials in average 2.8 ms ± 0.3 ms after the onset of the blue light pulse (see time-to- peak in Figure 7). EPSCs in V2a neurons were detected on average 3.7 ms ± 0.2 ms after the onset of the blue light pulse (see EPSC onset on Figure 7). This means that the lag between RB spiking and V2a EPSC was 0.9 ms on average, which is compatible with a monosynaptic connection from RB neurons onto V2a interneurons.

As a reference, in *Xenopus* tadpole, Wen-Chang Li in the Roberts lab used for double patch clamp recordings the criterion of 3 ms to distinguish mono- from polysynaptic currents (Li et al., 2007 Neural Development 2:17). In their conditions, monosynaptic connections identified by double patch clamp recordings were associated with delays ranging from 0.94 to 1.78 ms (mean, ± 0.10 ms, see Li et al., 2004 PNAS 101, 15488-493). From our study here on the connections from RB neurons onto ipsilateral V2a interneurons, we can estimate that the average axon distance to travel was approximately ~ 300 µm. Assuming a synaptic delay of 0.5 ms, the calculated conduction velocity was 0.75 m / s – a value compatible with previous reports in *Xenopus* tadpoles (Li et al.,2004 PNAS 101, 15488-493; Li et al.,2007 Neural Development 2:17).

We also added in the text a description of the EPSCs amplitude recorded in V2a interneurons, as well as the estimated depolarization induced in V2a cells (see lines subsection “Mechanosensory neurons synapse onto V2a interneurons involved in fast locomotion” in Results).

Our new results and analysis from physiology of the RB-V2a synapse are now detailed in the main text, subsection “Mechanosensory neurons synapse onto V2a interneurons involved in fast locomotion”. We added in Figure 7 the distribution of time-to-peaks for each RB cells and the EPSC delays for each V2a neurons. For each cell, we plotted the data’s standard deviation to illustrate the jitter.

Note that the jitter of EPSCs recorded in V2a neurons reflects both the variability in spike timing of the RB neurons in response to the optical stimulation as well as the variability in synaptic transmission from RB to V2a neurons. We therefore did not feel comfortable in using the jitter as an argument for monosynaptic connection, as it is used for double patch clamp recording where the precise timing of the presynaptic neuron is precisely known.

*We would like to be reassured that the nature of the connection is truly distinct from the one revealed by prior work, and is thus capable of influencing V2a activity during swimming.*

Previous work from the Roberts lab in *Xenopus* tadpole showed connections from RB neurons onto commissural excitatory interneurons (dIc and ecINs, see Li et al.,2007 Neural Development 2:17; Roberts et al.,2011 Developmental Neurobiology 575-584), which enable a delayed excitation onto contralateral dINs and motor neurons.

Here, we show for the first time that the dorsalmost *chx10+* V2a interneurons recruited during fast locomotion as shown previously (McLean et al.,2007 Nature 1;446(7131):71-5, McLean et al.,2008 Nature Neuroscience 11(12):1419-29) are receiving direct monosynaptic inputs from ipsilateral RB neurons. These connections from ipsilateral RB onto dINs were rarely observed by the group of Roberts (see Li et al.,2007 Neural Development 2:17), most likely because the synapses from RB onto ipsilateral dorsalmost V2a neurons are located dorsally at the midline, which is altered by sectioning for the open book tadpole preparation.

As described above, we have strong evidence that the projection we discovered using anatomy and optogenetics from RB neurons onto ipsilateral V2a dorsalmost interneurons is monosynaptic. Of course, we cannot exclude that there may be polysynaptic inputs onto V2a interneurons in addition to this monosynaptic component.

This result has important implication for the sensory modulation of motor circuits controlling speed in the spinal cord. As stated now in the Results section, EPSC amplitude induced by RB stimulation in V2a interneurons were on average 28.14 ± 3.9 pA, leading to estimated depolarization of 11.71 ± 2.59 mV on the V2a target (n = 8). The RB to V2a synapse is therefore relevant for determining spiking in V2a. For some V2a tested in current clamp mode, RB inputs were sufficient to elicit an action potential in V2a neurons (see example in Figure 7).

By this mean, RB neuron recruitment could therefore selectively feedback to interneurons driving high speeds and projecting onto a large pool of motor neurons downstream on the ipsilateral side. This point is now discussed in the Discussion, in light of results in *Xenopus* and lamprey.

Two important remarks on the relevance of this connection:

a) As mentioned previously, we cannot exclude that the behavioral effects we have described when silencing the output of *isl2b+* neurons with BotxLC-GFP could be due to connections from other sensory neurons targeted by the *Tg(isl2b:gal4)* line -- we therefore added to the revised manuscript a sentence to explain alternative interpretations of the behavior modulation via dorsal root ganglia and/or trigeminal neurons.

b) Interestingly, the connection from RB neurons onto ipsilateral V2a neurons is very selective for dorsal most V2a cells. When RB neurons are recruited at rest by a touch stimuli, an escape is triggered by initiation of a C bend on the contralateral side. This selective connection from RB onto dorsalmost V2a interneurons most likely is not sufficient to induce subsequent motor activity on the ipsilateral side at rest. However, this connection on the contrary could be modulatory and impact speed if RB neurons are recruited during active locomotion to modulate dorsalmost V2a spiking.

*The detailed reviews of two of the reviewers are included to help in revision but the focus of revision should be the two major points above.*

*Reviewer #1*

*1) The paper is well written within text but important details are missing from the legends that allow easy interpretation of the data. Some figures are too dense as if composed for a journal with restrictions on figure numbers.*

*All legends must be upgraded so that a reader can easily understand the data. Let me focus on Figure 1 to illustrate what must be done. What is going on in panels A and G? What are the superimposed lines in panel C (I know this but I am just saying be complete)? What is the color code in panel H and how does it relate to panel I. In panel I, I am really confused please explain the traces. Why are there multiple bumps in the left traces and what does the superposition mean in each set of traces? Please write figure legends that allow the data to be easily understood. Figure 3 is also a problem. What do the arrow and the arrowhead point out in panel A? What are we looking at in the left most image of panel C and in the middle image of panel C and left image of panel D? What are the different colors showing? In all graphs like panel E1 the mean and SEM are just too difficult to see especially amid the blue points; make bold.*

*Split Panels G to K of Figure 1 into a separate figure. Consider splitting Figure 5 also.*

We agree with the reviewer. Accordingly, we rewrote the legends to be more explicit. We also split Figure 1 into two figures, new Figure 1 and new Figure 2 by separating the GFP-Aequorin from the GCaMP data, as well as Figure 5, becoming new Figure 6 and new Figure 7 by separating the anatomy from the physiology.

*2) I am concerned about the morphological data shown in Figure 1. Was this observation made in a number of larvae? In Figure 5, was this observation made in a number of larvae. The legend of panel C implies that it was but you must be explicit. In Figure 5, was this observation made in a number of larvae?*

Since bioluminescence signals originating from single neurons are very small, we estimated activity of neuronal populations monitored by counting photons in a 1 ms bin on a single PMT (and the plotted them as number of photons per bin of 10ms. With this approach, single cell resolution is out-of-reach and specificity of the GFP-Aequorin expression pattern to the cells of interest is critical. Consequently, in the *Tg(mnx1:gal4;UAS:GFP-Aequorin-opt)* transgenic larvae, we had already previously verified that there was strictly no muscle expression by performing anti-GFP immunochemistry and complete larvae imaging on multiple larvae (n = 5, precision added in the main text, subsection “Mechanosensory neurons synapse onto V2a interneurons involved in fast locomotion” and in the Figure 1 legend, Video 1. In all animals tested (5 out of 5), there was strictly no muscle expression. We also added in the revised version the number of larvae analyzed for the old Figure 5 to B (i.e. new Figure 6 toB).

Similarly, note that the connectivity between RB neurons and V2a interneurons has been confirmed using anatomy and electrophysiology in all animals tested. For the anatomy in Figure 6 to B, we observed projections from RB neurons onto ipsilateral dorsalmost V2a neurons in 12 cells from 4 larvae. For the electrophysiology in Figure 7, we confirmed a monosynaptic connection in 8 out of from 8 recorded dorsal V2a neurons from 8 larvae that were selected based on the RB axonal projection observed onto V2a somata. Stimulating CoChR+ RB neurons led to reliable EPSCs with a short delay in all V2a neurons tested. Since the result was unambiguous and observed in all cells recorded based on anatomy, we estimated that these numbers were sufficient to conclude that a monosynaptic connection exists from RB sensory neurons onto dorsalmost V2a interneurons.

*3) There is no direct causal evidence that Rohon Beard cells do indeed mediate the speed changes associated with active swimming, because they are not selectively silenced or optogenetically activated. This needs to be more explicitly acknowledged.*

We agree with the reviewer. We had originally mentioned this limitation in the manuscript. As we cannot exclude that the behavioral effects we have described when silencing the output of *isl2b+* neurons with BotxLC-GFP could be due to connections from other sensory neurons targeted by the *Tg(isl2b:gal4)* line (see comments and references above), we added to the revised manuscript alternative interpretations of the behavior modulation via dorsal root ganglia or trigeminal neurons (see Abstract, Results Discussion).

*4) Given the central effects of pancuronium bromide why wasn't α-bungarotoxin used to immobilize the larvae in the experiments of Figure 2?*

Both α-bungarotoxin and pancuronium bromide paralyze animals by blocking the α 7 subunit of ionotropic cholinergic receptors. The concern that such drugs act on cholinergic receptors in the brain as well as on the periphery at the larval stage is as much valid for α- bungarotoxin than pancuronium bromide bath application (Prof. Yoichi Oda, *personal communication*).

Author response image 4.Amplitude of bioluminescence signals originating from motor neurons does not increase with the intensity of voltage applied to induce acoustic stimuli.(A) As shown here for 10 fish tested in control conditions and with pancuronium bromide, larger voltages actually led to decreased amplitude of the bioluminescence signals before paralysis (mean bioluminescence amplitudes across fish = 28.3 +/- 1.7; 25.7 +/- 4.6; 14.9 +/- 4.7 photons /10ms for stimuli voltage of respectively 1, 2 or 4 V, p = 0.03) and no correlation after paralysis. Furthermore, one can note that the stimulus voltage used in paralyzed conditions was equal or higher than in control conditions. (B) As shown here for 10 control fish and 10 immotile mutants, larger voltages did not lead to larger amplitude of the bioluminescence signals: there was no correlation in control siblings while immotile mutants showed smaller signals for larger stimuli (mean bioluminescence amplitudes across fish = 13.9 +/- 3.6; 10.5 +/- 6.4; 7.9 +/- 7.3; 5.0 =:- 2.0 photons /10ms for respective stimuli voltage of 2, 3, 5 or 8 V; p = 0.001). Furthermore, one can note that the stimulus voltage used in immotile mutants (mean = 3.9 +/- 0.1 V) was higher than in control conditions (mean = 2.4 +/- 0.01 V, p < 0.001).**DOI:**
http://dx.doi.org/10.7554/eLife.25260.020

*I am also concerned that you did not control for stimulus intensity in these experiments. In active fish and during pancuronium paralysis, is the size of the aequorin response dependent of stimulus amplitude? If it is, then comparisons of amplitudes across these two groups is suspect because different stimulus amplitudes were used.*

We thank the reviewer for pointing at this potentially confounding factor. In free-tailed animals, intensity of the auditory stimulus was gradually increased until it elicited an escape response in actively moving animals while in paralyzed animals, the intensity of the auditory stimulus was gradually increased until it elicited a bioluminescence signal. In immotile mutants, the intensity was set to the lowest value eliciting a bioluminescence signal. We checked that the amplitude of the bioluminescence signal was not correlated with the intensity of the acoustic stimulation (Figure 11). In the active versus paralyzed experiment, larger movements, and therefore signals, were actually obtained for *smaller* stimuli, whereas there was no correlation after paralysis (see Figure 11, legend). Note also that the voltage controlling the acoustic stimulus was higher in the paralyzed and immotile conditions compared to controls.

*Reviewer #3:*

*1) If RB neurons are encoding proprioceptive information in addition to the cutaneous exteroceptive information they are normally associated with, then activation of RB neurons during acoustic stimuli needs to be confirmed. The prediction is that RB neurons would respond to skin stimulation and acoustic stimulation when the tail can move, but not to acoustic stimuli when the tail is immotile. Without a better idea of whether RB neurons are in fact responding to self generated movements, then it is difficult to attribute the behavioral deficit to them and justify the ensuing circuit busting. For example, it could be that there are neurons in the DRG that are instead responsible.*

We agree with the reviewer. As explained above, we tried to perform calcium imaging with single cell resolution on RBs. Due to technical constraint, GCaMPs did not appear sensitive enough for detecting calcium transients associated with single spikes. Nonetheless, we performed GFP- Aequorin experiments using the *Isl2b* driver and observed signals only when the tail is motile, suggesting mechanosensory neurons (Rohon Beard, Trigeminal or DRG) are recruited during active locomotion. These experiments indicate that at least one cell type among RB, TG and DRG is recruited during active acoustic escapes --- but we cannot pin it down to one or a combination within the three cell types.

Nonetheless, we would like to point out that the ipsilateral connection RB => V2a neurons that we described here – motivated by the effect we saw on speed in the behavior dataset – is actually very interesting by itself:

a) it was never described by Li and Roberts, most likely for technical limitations associated with the open book preparation;

b) the RB-V2a synapse was associated with large EPSCs that could modulate spike timing in the V2a target (“EPSC amplitude induced by RB stimulation in V2a interneurons were on average 28.14 ± 3.9 pA, leading to estimated depolarization of 11.71 ± 2.59 mV on the V2a target (n = 8)”); c) this circuit may enable a speed selective modulation by projecting onto a subset of V2a neurons (the dorsal most ones), as the anatomy of RB neurons is not compatible with projections onto the dendrites of more ventral V2a neurons.

*2) There has been quite a bit of work in Xenopus tadpoles studying RB circuitry involved in cutaneous skin reflexes (e.g., Li et al., 2003, J Neurosci.; Li et al., 2004, J Neurophysiol). Abrupt stimulation of RB afferents generates a reflexive response away from the stimulus. Also, as I understand it, they demonstrate that drive from RB cells that potentially arises from movement-related stimulation would largely be filtered out by inhibition (Sillar and Roberts, 1988, Nature). However, RB neurons can reset or increase swimming rhythms when they are stimulated (e.g., Sillar and Roberts, 1992, J Neurosci), so there is clearly the potential for RB neurons to act as they are proposed to here. I think it would be worth explaining more how the findings here fit into previous work in tadpoles, as well as studies of dorsal cells (the RB equivalents) in lampreys (Buchanan and Cohen, 1982, J Neurophysiol; Christenson etal.,1988, Brain Res). I think the role of the RB-V2a connection would be worth considering in the context of struggling as well (Li et al., 2007, J Neurosci).*

We agree with the reviewer that this discussion is important and was missing from the original submission. We revised the Discussion accordingly, see new paragraph in Discussion of revised manuscript, see below:

“Previous studies have described a remarkably simple and conserved disynaptic cutaneous skin reflex relying on spinal mechanosensory neurons referred to as Rohon Beard neurons in *Xenopus* and zebrafish and “dorsal cells” in lamprey (Buchanan and Cohen, 1982; Christenson et al., 1988; Clarke et al., 1984; Easley-Neal et al., 2013; Li et al., 2003; Roberts et al., 1983). […]. While local reflex pathways have been characterized at rest, further studies will decipher the mechanisms underlying the diversity of behavioral effects of stimulating mechanosensory neurons: escapes for single touch at rest (Buchanan and Cohen, 1982; Douglass et al., 2008; Roberts et al., 1983), struggling for continuous stimulation (Li et al., 2007b) and enhancement of speed during active locomotion (this study).”

*3) The optogenetic-electrophysiology data as currently presented don't provide sufficient temporal resolution to determine if the RB-V2a connection is monosynaptic or not. If it is, this would be novel, since studies of RB circuits in Xenopus have only revealed indirect connections from RBs to V2a-like neurons (with extremely short latencies though). In fact, the responses have multiple components which would be more consistent with a polysynaptic connection (especially since RB neurons tend to fire once). Given this is not the main goal of the work, I would suggest perhaps stepping back from saying it is a direct connection and instead focus on the excitatory nature of that connection, monosynaptic or not. Of course, this could all be moot if the RBs are not even active during body bending.*

We thank the reviewer for pointing at this critical aspect. As stated above, we analyzed the synaptic connectivity from RB neurons in 4 additional V2a interneurons, reaching a total of 8 cells.

To demonstrate the monosynaptic nature of the RB-V2a connection we analyzed the time-to- peak of elicited action potentials in CoChR-tdtomato^+^ RB neurons in response to 5 ms blue light stimulations (103 stimulations in 6 cells) as well as the delay of ESPCs in *chx10^+^*V2a neurons following RB stimulations (78 EPSCs in 8 cells).

Our analysis shows that RB fire single action potentials in average 2.80 ± 0.29 ms after the onset of the blue light pulse. EPSCs in V2a neurons were detected 3.71 ± 0.19 ms after the onset of the blue light pulse. This means that, in average, there is a lag of 0.9 ms between the peak of RB spikes and V2a EPSCs, which is compatible with a monosynaptic component. See details on the jitter for each cell in Figure 7, subsection “Mechanosensory neurons synapse onto V2a interneurons involved in fast locomotion” and Discussion above.

*4) For the botulinum experiments it is largely assumed that the sensory populations are silenced, but it would be better if there was some confirmation in this line of fish. Patch recordings would be ideal, but if not possible a mention of whether the fish fail to respond to touch or not would suffice.*

We previously showed evidence that BoTxBLC-GFP was effective at silencing synaptic release in motor neurons, GABAergic sensory neurons as well as glutamatergic V2a neurons (Sternberg et al., Current Biology 2016). In one case, we could perform the proper demonstration that silencing of GABA release was fully effective by recording the postsynaptic target while stimulating the presynaptic GABAergic sensory neurons using optogenetics. This demonstration was achieved in triple transgenic animals Tg(pkd2l1:gal4;*UAS:BoTxBLC-GFP; UAS:ChR- mCherry*) where the targets of GABAergic sensory neurons, CaP motor neurons, is obvious due to the basket structure formed by the axons of the GABAergic cells (see Hubbard *et al.,* Current Biology 2016; Sternberg *et al., Current* Biology 2016).

However, to demonstrate full blockade of synaptic release from mechanosensory neurons, we would need to have the opsin CoChR-tdTomato combined with the BotxBLC-GFP in RB neurons expressing Gal4, and a fluorescent tag in one of RB targets, such as CoPA or dorsalmost V2a neurons. This experiment would require to inject the construct (UAS: CoChR-tdTomato) in triple transgenic Tg(*isl2b:gal4;UAS:BoTxLC-GFP; Chx10:DsRed*) eggs. Unfortunately, we did not raise triple transgenic animals with this combination of transgenes and could not therefore attempt this experiment within few months.

In our silencing experiments, we have indications that in the *Tg(isl2b:gal4;UAS:BoTxLC-GFP)* double transgenic larvae only a fraction of mechanosensory neurons are silenced:

a) the *Tg(isl2b:gal4;UAS:BoTxBLC-GFP)* transgenic larvae do not show GFP in all RB neurons: due to variegation of the UAS/Gal4, only a fraction (between 1/3^rd^ to 2/3^rd^ depending on the fish) of the total number of RB neurons are labeled;

b) 2 dpf *Tg(isl2b:gal4;UAS:BoTxBLC-GFP)* transgenic larvae larvae upon touch stimulation respond in about half the cases – which is compatible with our point a). We added a comment on the effectiveness of our silencing method in the manuscript’s Result section, see below:

“Due to the fact that UAS/Gal4 labels cells in a variegated manner, we only observed BotxLCB- GFP in half the RB neurons so although the isl2b promoter drives expression in all RB neurons, dorsal root ganglia and trigeminal neurons. This suggests that the effect of silencing mechanosensory feedback in our behavior experiments are underestimated in our experiments.”

*5) Since the RB driven response is specific to high frequency swimming commensurate with large deflections of the tail, I was expecting that the controls for the aequorin experiments would be zebrafish that are fully embedded so the tail can't move. Is it known if there is still less motor neuron recruitment in this condition comparable to toxin and mutants?*

From our previous observations during the study of GABAergic sensory neurons in the spinal cord (Böhm et al., Nature Communications 2016), non paralyzed larvae that are embedded in agar perform very strong local contractions locally, which look like transient “accordion” where the body is firmly contracting against the agar wall. During these local contractions of the tail confined in agar, there is most likely stimulation of RB neurons via friction against the agar. We therefore did not consider that this experimental paradigm would constitute a good negative control.

*6) Work in lampreys studying the role of edge-cells seems appropriate to discuss more, since the function appears to be very similar to that proposed here and a lot of that work is performed while the tail is moving. Also the recent development of a rodent preparation is another precedence for neural recordings in moving animals (Hayes etal., 2009, J Neurophysiol; Hochman et al., 2012, Front. Biosci).*

We agree and added references to the extensive work performed in lamprey on edge cells in preparations where the tail is free in particular, see Discussion.

[Editors' note: further revisions were requested prior to acceptance, as described below.]

*1) There was concern that the authors overstated the case for monosynapticity in the Introduction and Results, subsection “Mechanosensory neurons synapse onto V2a interneurons involved in fast locomotion****”****, given that experiments with a high concentration of divalent ions – or the like – to eliminate (reduce the probability of) disynaptic events were not performed. We suggest that the text starting in subsection “Mechanosensory neurons synapse onto V2a interneurons involved in fast locomotion****”***
*be modified something like "To assess whether RB-V2A ipsilateral connection might be monosynaptic, we analyzed the lag between RB spikes time-to-peak and V2a EPSCs onset and found that it was below 1 ms (Figure 7, 0.9 ms), a value compatible with monosynaptic connection (see (Li et al., 2007a) where cutoff is set at 3 ms for monosynaptic connections)." The Introduction can be correspondingly edited.*

We edited the corresponding lines as suggested:

Summary: *In the spinal cord, we show that connections compatible with monosynaptic inputs from mechanosensory Rohon-Beard neurons onto ipsilateral V2a interneurons selectively recruited at high speed can contribute to the observed enhancement of speed.*

Subsection “Mechanosensory neurons synapse onto V2a interneurons involved in fast locomotion”: *To assess whether RB-V2A ipsilateral connection might be monosynaptic, we analyzed the lag between RB spikes time-to-peak and V2a EPSCs onset and found that it was below 1 ms (Figure 7, 0.9 ms), a value compatible with monosynaptic connection (see (Li et al., 2007a) where cutoff is set at 3 ms for monosynaptic connections).*

*2) Quoting one of the expert reviewers "There is an underlying assumption that all UAS lines (GFP-aequorin-opt and GCaMP6f;cryaa:mCherry) will produce the same expression patterns when paired with this Gal4 (mnx1:gal4) line. Just because the icm23 line (mnx1:gal4-UAS:preEGFP) was characterized by Bohm et al., this does not mean that Gal4 will work the same with different versions of UAS (UAS:GFP-aequorin-opt or GCaMP6f;cryaa:mCherry). Have these lines been characterized? If so, references to that work must be made to show MN specificity/variegated expression patterns (if any)."*

So far, we only observed motor neurons when we used the mnx1 promoter to drive expression of GCaMP5G (Sternberg et al., Current Biology 2016) or Gal4 in a UAS:RFP line. However, we cannot exclude that other spinal neurons, distinct from motor neurons, may be targeted by this promoter. Therefore we added in the Material and methods the following lines:

Given the potential variability in expression patterns of Gal4 lines with different UAS, we characterized the Tg(mnx1:gal4;UAS:GCaMP6f,cryaa:mCherry) line and observed a similar expression in motor neurons as in the Tg(mnx1:gal4;UAS:GFP-aequorin-opt)^icm09^ line. In both cases, we observed the targeting of spinal neurons exiting the spinal cord in the ventral root, and therefore referred to as motor neurons. However, we cannot exclude that the mnx1 promoter may target other spinal neurons, distinct from motor neurons.